# RNA N6-methyladenosine modification-based biomarkers for absorbed ionizing radiation dose estimation

Hongxia Chen [1,7], Xi Zhao[1,7], Wei Yang[2,7], Qi Zhang[1,3], Rongjiao Hao[1,4], Siao Jiang[1,4], Huihui Han[1], Zuyin Yu[1], Shuang Xing[1], Changjiang Feng[5], Qianqian Wang[2], Hao Lu[1], Yuanfeng Li[1], Cheng Quan[1], Yiming Lu [1,4] ✉ & Gangqiao Zhou [1,3,4,6] ✉

Radiation triage and biological dosimetry are critical for the medical management of massive potentially exposed individuals following radiological accidents. Here, we performed a genome-wide screening of radiation-responding mRNAs, whose N6-methyladenosine (m⁶A) levels showed significant alteration after acute irradiation. The m⁶A levels of three genes, *Ncoa4*, *Ate1* and *Fgf22*, in peripheral blood mononuclear cells (PBMCs) of mice showed excellent dose-response relationships and could serve as biomarkers of radiation exposure. Especially, the RNA m⁶A of *Ncoa4* maintained a high level as long as 28 days after irradiation. We demonstrated its responsive specificity to radiation, conservation across the mice, monkeys and humans, and the dose-response relationship in PBMCs from cancer patients receiving radiation therapy. Finally, *NOCA4* m⁶A-based biodosimetric models were constructed for estimating absorbed radiation doses in mice or humans. Collectively, this study demonstrated the potential feasibility of RNA m⁶A in radiation accidents management and clinical applications.

With the widespread application of nuclear technology, including nuclear medicine and nuclear power, more attentions should be paid to the development of high-throughput methods for screening and diagnosing the exposed individuals in a mass casualty radiological incident. High dose of ionizing radiation (IR) (usually >1 Gy gamma ray) can lead to acute radiation syndrome (ARS), which involves multiple organ systems, with symptoms ranging from mild ones such as nausea and vomiting, to death[1,2]. According to distinct clinical outcomes, human ARS can be generally classified into three major subsyndromes, tightly related to the absorbed dosage, including hematopoietic (2–6 Gy), gastrointestinal (6–10 Gy) and neurovascular (>10 Gy)

ones[2–4]. An accurate and efficient radiation dose assessment is absolutely necessary to triage IR-exposed victims into definable, treatment-susceptible groups.

In the past decades, biological dosimetry methods such as the micronucleus assay and dicentric assay have been used for real-life exposure cases; however, these methods still have limitations in analyzing large number of samples, particularly due to the ~48 hours (h) minimum culture time required prior to analysis. Gamma-H2AX (γ-H2AX), a classical biomarker of DNA double-strand breaks (DSB), is also considered as a rapid and sensitive radiation biomarker. Additionally, the gene expression signatures have been explored for the

[1]State Key Laboratory of Proteomics, National Center for Protein Sciences at Beijing, Beijing Proteome Research Center, Beijing Institute of Radiation Medicine, Beijing, China. [2]Department of Radiation Oncology, the First Medical Center of Chinese PLA General Hospital, Beijing, China. [3]School of Medicine, University of South China, Hengyang City, Hunan Province, China. [4]School of Life Science, University of Hebei, Baoding City, Hebei Province, China. [5]Department of Thoracic Surgery, the First Medical Center of Chinese PLA General Hospital, Beijing, China. [6]Collaborative Innovation Center for Personalized Cancer Medicine, Center for Global Health, School of Public Health, Nanjing Medical University, Nanjing City, Jiangsu Province, China. [7]These authors contributed equally: Hongxia Chen, Xi Zhao, Wei Yang. ✉e-mail: luymnet@126.com; zhougq114@126.com

prediction of irradiation doses[5–9]. For example, a collection of DNA damage repair-related genes have been reported to be potential biomarkers for radiation exposure, including *DDB2*, *XPE* and *XPC*[10–12]. Recently, a study reported the serum microRNAs miR-150-5p as a potential radiation biodosimeter in mice and leukemia patients underwent radiotherapy[13,14]. However, the alterations of these molecules expression post irradiation are highly dynamic and usually decay rapidly after exposure, so the valid time window for detection is limited[15]. On the contrary, DNA methylations are highly stable under various stresses and have been identified as biomarkers for multiple diseases[12,16–18]. However, their slow responses to stresses make them unsuitable for the rapid detection of radiation. Therefore, a class of molecules with a good balance between responsive dynamics and biological stability upon stress might be served as more preferred biomarkers for irradiation exposure.

RNA molecules play important roles in biology, and post-transcriptional modifications of RNA in cells play crucial roles in the regulation of its stability, transport, processing, and gene expression[19]. Among more than 170 types of RNA modification found so far, N[6]-methyladenosine (m[6]A) is one of the most common and abundant RNA modifications and has been shown widely involving in a variety of biological and disease processes[15,20–23]. Recently, several studies have reported that RNA m[6]A modification was involved in DNA damage repair to combat single-strand breaks (SSBs) induced by ultraviolet (UV) and DSBs induced by IR and chemical agents[24–26], possibly by regulating the stabilization/destabilization of R-loops at DSBs[27]. Given their biological and pathological importance, m[6]A enzymes have been investigated as potential biomarkers for disease prognosis, especially for cancers[28–30]. However, up to now, there are still lack of studies on the direct application of RNA modifications as biomarkers, not limited to irradiation biomarkers. Thus, it is of worth to explore m[6]A-related mRNAs in response to IR and determine its feasibility as biomarkers for assessment of radiation exposure dose.

Here, we performed a genome-wide screening for mRNA transcripts whose m[6]A modification levels reveal significant changes after IR, and identified the m[6]A modifications in transcripts of three genes, *Ncoa4*, *Ate1* and *Fgf22*, as potential biomarkers for IR. Further, we selected the m[6]A modification in *Ncoa4* as an example, and examined its dose-response relationship, temporal dynamics, stress specificity and cross-species conservation, and assessed its potential clinical utility in patients receiving radiotherapy. Finally, we seek to construct biodosimetric models for estimating the absorbed radiation doses based on *NOCA4* m[6]A modification. Our study demonstrated the potential feasibility of using RNA m[6]A modification level in radiation accident management and clinical application.

## Results

### Construction of total-body irradiation mice models

In order to systemically discover the candidate RNA m[6]A biomarkers responding to irradiation exposure, we constructed a set of adult mice total-body irradiation (TBI) models exposed to different dosage of gamma rays (0.2, 0.5, 1, 2, 4 and 6.5 Gy). Mice in control group were sham-exposed. Physiological phenotypes of mice in each group were measured at multiple time points within 28 days following irradiation. Our results showed that the body weights of mice exposed to gamma rays ≥ 0.5 Gy significantly decrease within the first 3 days after irradiation and then increase gradually, and their body weights are lower than the mice in control group through the entire 28-day time window post irradiation (Fig. 1a). Hemogram analyses of the murine peripheral whole blood showed that the counts of white blood cell (WBC), lymphocyte (Lym) and monocyte (Mono) are markedly reduced at the first day post irradiation in a dose-dependent manner (Supplementary Fig. 1a–c). For the WBC and Lym counts, their reductions lasted for at least 14 days after irradiation for mice exposed to a dose ≥ 2 Gy; while for the Mono counts, the apparent reduction lasted for at least 14 days

for mice exposed to a dose of ≥ 4 Gy, but lasted for less than 3 days for mice exposed to a dose ≤ 2 Gy. Comparably, the decrease of the counts of red blood cell (RBC), hemoglobin (HGB) and platelets (PLT) after IR was relatively moderate, especially for mice exposed to a dose ≤ 2 Gy. For mice exposed to a dose ≥ 4 Gy, apparent reductions of the RBC and HGB counts appeared at day 14 after irradiation and reduction of the PLT counts appeared at day 7 (Supplementary Fig. 1d–f). Notably, the counts of WBC and Lym in mice exposed to higher radiation dose (*e.g.*, 6.5 Gy) failed to fully recover even at day 28 post IR, which were consistent with previous studies[31].

### Two-stage transcriptome-wide screening of candidate RNA m[6]A biomarkers responding to irradiation in mice PBMCs

Next, we performed a two-stage profiling of transcriptome-wide RNA m[6]A levels and RNA expression levels in peripheral blood mononuclear cells (PBMCs) from the TBI mice using the Mouse RNA Epitranscriptomic Microarray (8 × 60 K, Arraystar, Rockville, MD, USA) (see Methods). In stage I, we focused on revealing the temporal dynamics of RNA m[6]A after irradiation and identifying the RNA transcripts with constantly altered m[6]A levels during a relatively long period of time post irradiation (TPI) in mice PBMCs. In this stage, PBMCs were collected from the sham-exposed mice and mice exposed to a dose of 6.5 Gy gamma ray at five different time points (day 1, 3, 7, 14, and 28 post irradiation), and the transcriptome-wide RNA m[6]A modification levels and RNA expression levels in these samples were obtained (Fig. 1a, see Methods). In stage II, we aimed to further screen RNA transcripts that still exhibit significantly altered m[6]A levels when mice were exposed to a lower dose. In this stage, PBMCs were collected from the sham-exposed mice and mice exposed to a dose of 2 Gy gamma ray at two key time points (day 1 and 14 post irradiation) for the quantification of the transcriptome-wide RNA m[6]A modification levels and RNA expression levels (Fig. 1b).

Then, the differentially expressed and m[6]A methylated genes were identified, respectively, at each time point post irradiation in the stage I (see Methods). We observed that generally the RNA m[6]A levels reveal lower dynamics than the expression profiles in PBMCs after irradiation, as fewer number of RNA exhibited altered m[6]A levels than altered expression levels across different time points (Fig. 1c, d). Specifically, the gene expression profiles showed dramatic changes across all the time points, with more genes being down-regulated than being up-regulated. In contrast, RNA m[6]A profiles exhibited relatively larger changes at day 3 and day 14 than at the other time points, with significantly higher number of genes showing hyper- and hypo-methylation, respectively.

To explore the temporal patterns of m[6]A across different time points post irradiation, we first divided RNA transcripts into two broad categories, "Hyper" and "Hypo", based on whether they were hyper- or hypo-m[6]A methylated at one or more time points. The inconsistent transcripts with both hyper- and hypo-methylation at different time points were removed from further analyses. Then, using an unsupervised clustering method, we showed that RNAs in "Hyper" or "Hypo" category can be respectively divided into two sub-categories (Fig. 1e, f). Among the two hyper-methylation sub-categories, one sub-category contains consistently hyper-methylated transcripts across all the time points, and the other sub-category contains transcripts showing hyper-methylation only at day 3 post irradiation. Similarly, among the two hypo-methylation sub-categories, one contains consistently hypo-methylated transcripts, and the other one contains transcripts showing hypo-methylation only at day 14 post irradiation. Comparing with transcripts in the other three sub-categories, the transcripts in the consistent hyper-methylation sub-category are more suitable candidates serving as the biomarkers responding to irradiation. Functional annotation of genes related to m[6]A-methylated RNAs in the four sub-categories showed that they are associated to different biological processes (Supplementary Fig. 2a–d). For examples, genes

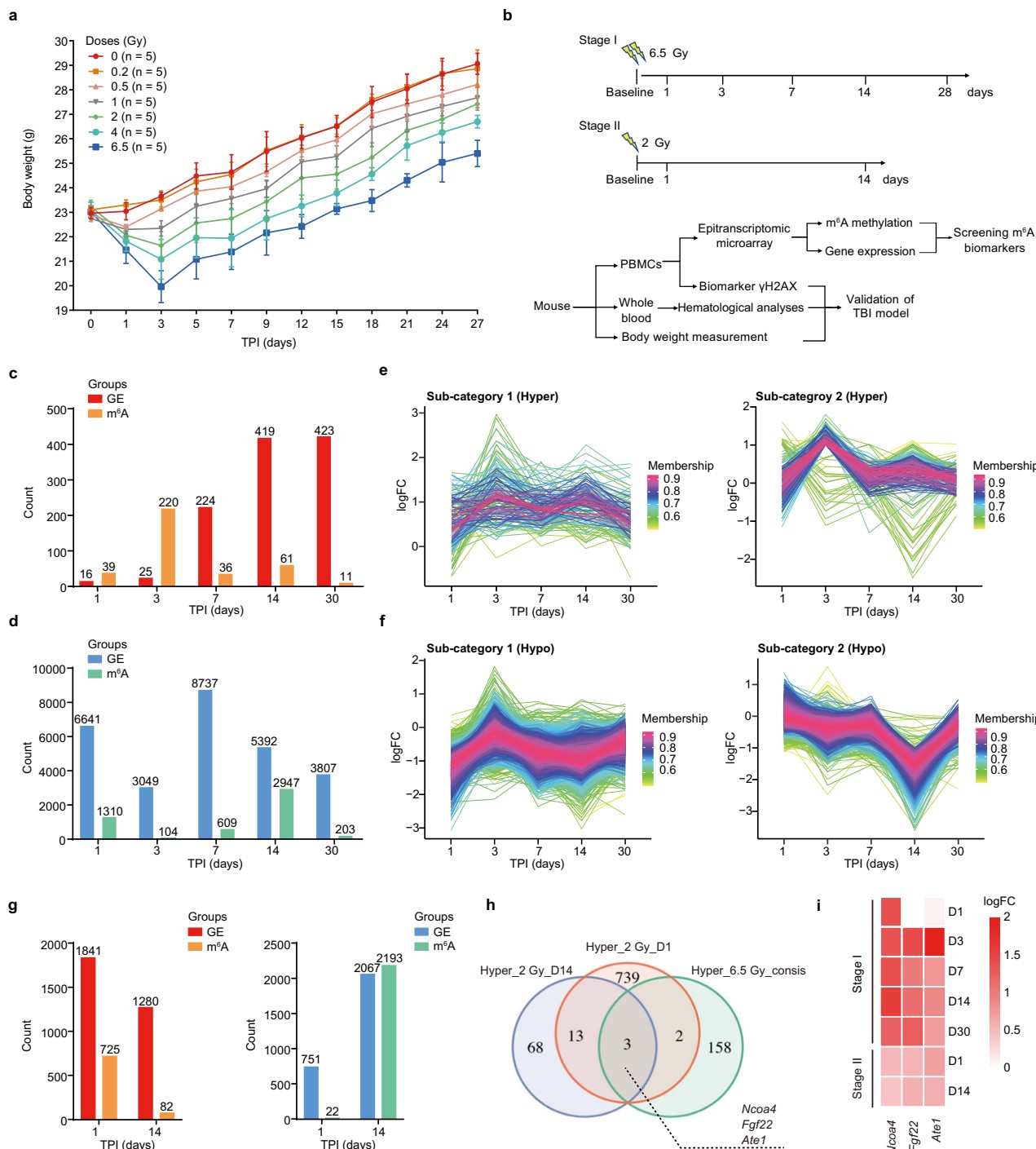

**Fig. 1 | Dynamic changes in the transcriptomes and epi-transcriptomes of PBMCs in irradiated mice. a** Body weight of 6–8 weeks old C57BL/6 male mice after total-body irradiation (TBI) by gamma rays at different doses ($n = 5$ for each time point). Data are presented as means ± standard deviation (SD). **b** Schematic for a two-stage screening of N6-methyladenosine (m6A) modification responsive to gamma-ray radiation in peripheral blood mononuclear cells (PBMCs) from the mice exposed to gamma-ray radiation ($n = 5$ for each time point). **c** The significantly up-regulated genes and hyper-m6A methylated genes in PBMCs from mice at multiple time points (days 1, 3, 7, 14, and 30) after 6.5 Gy gamma-ray radiation. GE: gene expression profile. **d** The significantly down-regulated genes and hypo-m6A methylated genes in PBMCs from mice at multiple time points (days 1, 3, 7, 14, and 30) after 6.5 Gy gamma-ray radiation. The sub-categories of hyper- (**e**) and hypo-methylated (**f**) RNAs identified by unsupervised clustering analyses. X axis represents the time post irradiation, Y axis represents the log-transformed fold changes

of m6A levels. Each line represents the dynamic changes of the m6A levels of each RNA transcript, and the membership of lines is represented by colors. **g** The significantly up-regulated genes and hyper-m6A methylated genes (left panel) and down-regulated genes and hypo-m6A methylated genes (right panel) in response to 2 Gy gamma-ray radiation at two time points (day 1 and 14) in mice PBMCs. **h** Venn diagram showing the overlaps among three RNA sets: consistently hyper-m6A methylated RNAs after 6.5 Gy irradiation (Hyper_6.5Gy_consis), hyper-m6A methylated RNAs after 2 Gy irradiation at day 1 (Hyper_2Gy_D1), and hyper-m6A methylated RNAs after 2 Gy irradiation at day 14 (Hyper_2Gy_D14). **i** A heat map showing the m6A modification levels of *Ncoa4*, *Ate1*, and *Fgf22* mRNAs across all the time points in stages I (days 1, 3, 7, 14, and 30) and II (days 1 and 14) experiments. Colors represent logFC of m6A levels between PBMCs from the irradiated mice and control mice. Source data are provided.

related to the consistent hyper-methylation sub-category are mainly involved in regulation of blood pressure and developmental growth (Supplementary Fig. 2a), while genes related to consistent hypo-methylation sub-category are mainly related to covalent chromatin modification and DNA repair (Supplementary Fig. 2c).

We then analyzed the RNA expression and m$^6$A methylation profiles of PBMCs from the TBI mice in stage II, who were exposed to a lower dose of 2 Gy gamma ray. Similar to stage I, less number of RNA transcripts showed altered m$^6$A levels than altered expression profiles. Specifically, a total of 757 and 84 RNA transcripts were significantly hyper-methylated at day 1 and day 14 post-irradiation, respectively (Fig. 1g). To robustly identify the biomarkers responding to irradiation exposure, we performed overlapping analysis among the transcripts in the consistent hyper-methylation sub-category in stage I and transcripts hyper-methylated at day 1 or day 14 post irradiation in stage II. Finally, we identified three transcripts, which were related to genes *Ncoa4*, *Ate1*, and *Fgf22*, simultaneously presenting in all three sets and showed consistent hyper-methylation across all the time points in stages I and II (Fig. 1h, i).

### Validation of the temporal responding patterns of m$^6$A in *Ncoa4*, *Ate1*, and *Fgf22* after irradiation

Next, we sought to validate the temporal dynamics of m$^6$A modification at *Ncoa4*, *Ate1*, and *Fgf22* transcripts in response to IR using the methylated RNA immunoprecipitation in combination with real-time quantitative polymerase chain reaction (MeRIP-qPCR) assays. Because the m$^6$A microarray detects m$^6$A modification at the transcript level, the locations of those methylated adenines in the transcripts must be first determined before MeRIP-qPCR primers could be designed. RNA m$^6$A sites prediction tool SRAMP[32] was then used to predict the highly confident m$^6$A sites distributed across *Ncoa4*, *Ate1*, and *Fgf22* transcripts (Supplementary Fig. 3a–c). Then, the primers for MeRIP-qPCR assays were designed to target highly confident m$^6$A sites in *Ncoa4*, *Ate1*, and *Fgf22* transcripts after separating them into regions of 100 - 200 base pairs (bp) length (Fig. 2a–c, Supplementary Data 1). After primer feasibility evaluation (Methods), the primers targeting the m$^6$A sites of *Ncoa4* mRNA (primer 1 for A459; and primer 2 for A761, A781 and A799), the m$^6$A sites of *Ate1* mRNA (primer 2 for A767; and primer 3 for A1782) and the m$^6$A sites of *Fgf22* mRNA (primer 2 for A370) were selected for quantifying their m$^6$A levels by MeRIP-qPCR assay (Fig. 2d–f). Consistent with the microarray data, the MeRIP-qPCR assays showed that the RNA m$^6$A levels of these three genes were significantly up-regulated after irradiation (Fig. 2d–f).

We then quantified the RNA m$^6$A methylation levels of *Ncoa4*, *Ate1*, and *Fgf22* in PBMCs collected from mice at multiple time points (day 1, 3, 7, 14, and 28) after exposure to varying doses (0.2, 0.5, 1, 2, 4, 6.5, and 10 Gy) of gamma rays TBI. Due to the high mortality rate of mice shortly after exposure to 10 Gy TBI, their PBMCs were only collected at two time points (day 1 and 3) after irradiation. We observed that the curves of *Ncoa4* mRNAs m$^6$A levels across different time points were elevated significantly after irradiation, and the extend of elevation was clearly related to exposure dosage either using *Ncoa4* primer 1 or primer 2 (Fig. 2g and Supplementary Fig. 3d; Supplementary Data 2). Notably, with exposure doses ≥2 Gy, the m$^6$A levels of *Ncoa4* peaked at day 14 and could last for at least 28 days after irradiation (Fig. 2g and Supplementary Fig. 3d; Supplementary Data 2). Moreover, we found the elevation of *Ncoa4* mRNA m$^6$A levels are still distinguishable with a lower dose (0.5 or 1 Gy) at day 1 or 3 after irradiation. Together, these results indicate that the *Ncoa4* mRNA m$^6$A modification could serve as a candidate irradiation biomarker, which is suitable for short-period (1 day) detection after irradiation at doses ≥0.5 Gy, but also for long-period (28 days) detection after irradiation at doses ≥2 Gy.

Similar to *Ncoa4* mRNAs, the curves of *Ate1* and *Fgf22* mRNAs m$^6$A levels across different time points were elevated significantly after irradiation with clear association to exposure dosage using either *Ate1*

primer 2 or primer 3, and *Fgf22* primer 2, respectively (Fig. 2h,i and Supplementary Fig. 3e; Supplementary Data 2). However, the m$^6$A levels of the two transcripts peaked at day 1 and the duration of elevation was considerably shorter than those of *Ncoa4*. Notably, the recovery duration of m$^6$A levels back to the baseline is associated to the radiation dose. At a dose of 6.5 Gy, the m$^6$A of *Ate1* mRNA is continuously detectable within 28 days after irradiation (Fig. 2h). With irradiation doses <1 Gy, the m$^6$A levels of *Ate1* returned to the baseline at day 7 after irradiation, and for *Fgf22* transcripts, the recovery duration was even shorter (Fig. 2i). Notably, the expression levels of *Ncoa4*, *Ate1*, and *Fgf22* mRNAs did not show significant differences after irradiation along with the time and dose (Fig. 2j and Supplementary Fig. 3f, g). Thus, the effects of mRNA expression changes of these three genes on their m$^6$A levels can be largely excluded. Together, these results indicate that the m$^6$A modification of *Ate1* and *Fgf22* mRNAs can be used as irradiation biomarkers in the early stage after irradiation, with the most significant increase over a short period of time (1 day) and can be applied in longer post-irradiation time (7 days) scenarios when the irradiation dose is high (≥ 4 Gy).

In addition to MeRIP-qPCR, we adopted a single-base and non-antibody-based m$^6$A mapping method SELECT[33] to further assess the responding patterns of m$^6$A at *Ncoa4* transcripts upon IR. Using SELECT probe pairs targeting the highly confident A459, A761, A781 and A799 sites at *Ncoa4* transcript respectively, we measured their m$^6$A statuses in the PBMCs from mice at 7 and 14 days after 1, 2, 4 and 6.5 Gy gamma rays TBI. We observed that, three (A459, A761 and A781) of the four targeted sites reveal significantly elevated m$^6$A levels at both 7 and 14 days after irradiation, and the extend of elevation is clearly related to exposure dosage (Fig. 2k, l and Supplementary Fig. 4a, b). This results thus confirm the results of MeRIP-qPCR on the responding curves of *Ncoa4* m$^6$A levels upon IR.

We also compared the temporal responding curves of m$^6$A modification of *Ncoa4*, *Ate1* and *Fgf22* mRNAs with a classical irradiation biomarker γ-H2AX. Immunofluorescence results showed that the increase of γ-H2AX foci is almost undetectable at doses ≤1 Gy at day 1 or later after irradiation (Supplementary Fig. 5a, b; Supplementary Data 2). Even with an exposure dose of 6.5 Gy, the γ-H2AX foci became undetectable at day 7 after irradiation. Consistently, γ-H2AX protein could only be detected within 1 day after exposure to a dose ≥2 Gy (Supplementary Fig. 5c; Supplementary Data 2). These data demonstrate that the m$^6$A modifications of *Ncoa4*, *Ate1*, and *Fgf22* mRNAs outperform the classical radiation biomarker γ-H2AX both in the detection sensitivity of low dose irradiation and long detectable time window after exposure. Among these three candidate genes, the m$^6$A levels of *Ncoa4* mRNA showed the most significant changes and the longest duration post-irradiation, making it the most ideal candidate for an irradiation biodosimeters applying in dose prediction at lower dose and longer time range. Therefore, we focused on the m$^6$A modification of *Ncoa4* mRNA in subsequent studies.

### Response of *NCOA4* m$^6$A modification to radiation exposure in PBMCs of non-human primates (NHPs) and human cells

To assess the response of *NCOA4* m6A modification to radiation exposure in NHPs, we first performed a conservation analysis of the sequences surrounding the m$^6$A sites among the homologous genes of *Ncoa4* in multiple species. We found that the four highly confident m$^6$A sites (A459, A761, A781 and A799) in mouse *Ncoa4* mRNA are highly conserved across species (Fig. 3a). Prediction of the m$^6$A sites at *NCOA4* mRNA in monkeys and humans by SRAMP[32] also showed the four m$^6$A sites are of high confidence (Fig. 3b, c and Supplementary Fig. 6a, b). We then re-designed three primer pairs to target the four conserved m$^6$A sites at monkey and human *NCOA4* mRNA, respectively. After primer feasibility evaluation, the primers targeting the m$^6$A sites (primer 2 for A896 and A914; and primer 3 for A1229) in monkey *NCOA4* mRNA and the m$^6$A sites (primer 2 for A886 and A924; and

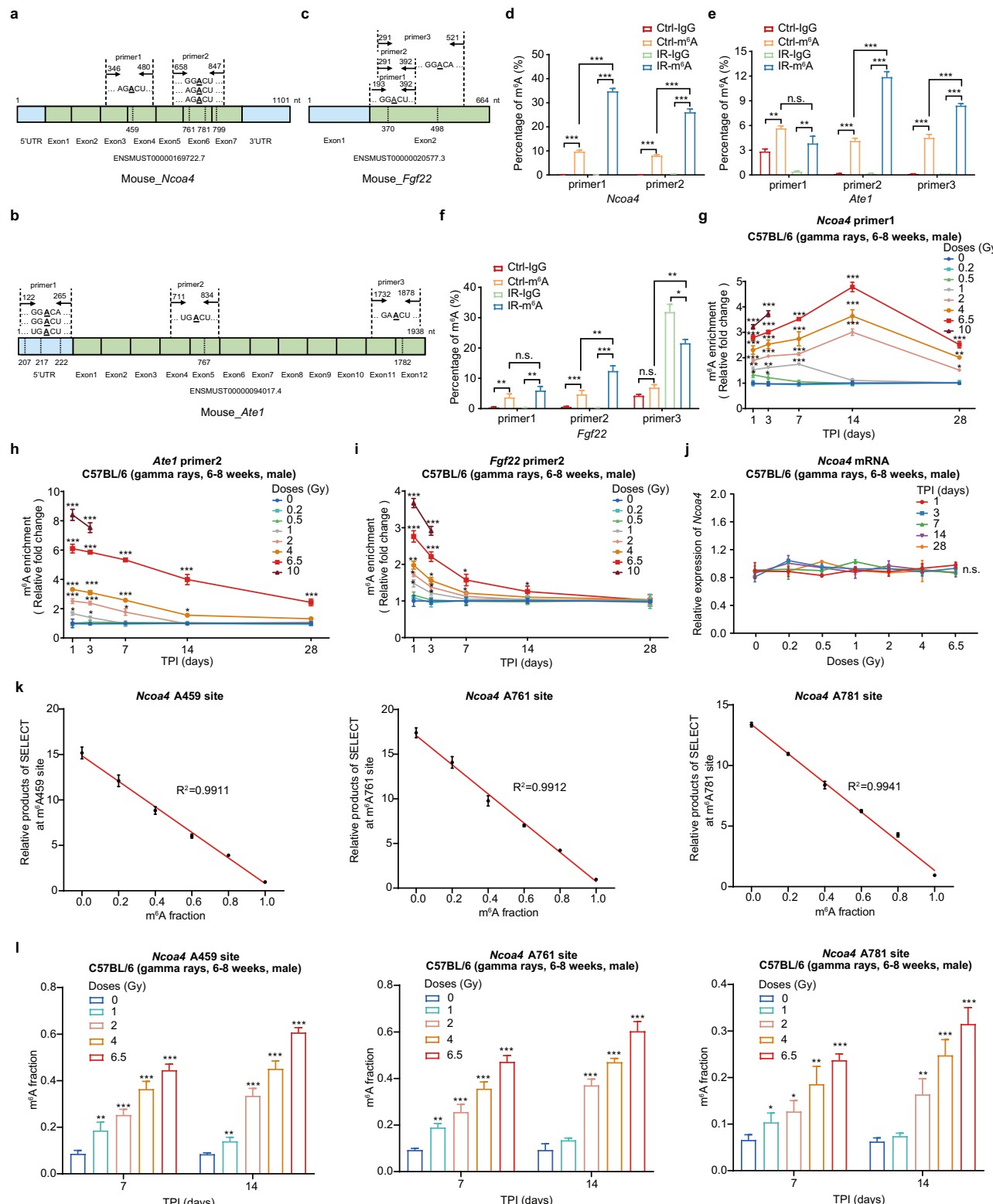

primer 3 for A1239) in human *NCOA4* mRNA were selected for quantifying the m⁶A levels by MeRIP-qPCR assay (Fig. 3d,e).

To assess the potential utility of *NCOA4* m6A modification for dose estimation in NHPs, we established a rhesus monkey (*Macaca mulatta*) gamma-ray TBI model (*n* = 6) using a moderate dose (6.75 Gy) of irradiation as previously described[34] (see Methods). PBMCs were isolated at one day before irradiation and 5 time points post-irradiation (days 1, 7, 14, 21, and 28). The MeRIP-qPCR results showed that the m⁶A levels of *NCOA4* mRNA from monkeys PBMCs increase significantly

after irradiation and peak at day 14 post-irradiation using either *NCOA4* primer 2 or primer 3, similar to the responding curves observed in the TBI mice models with a dose of 6.5 Gy (Fig. 3f).

Next, we examined the responding curves of *NCOA4* m⁶A in human umbilical vein endothelial cells (HUVECs) after irradiation. The HUVECs were exposed to varying doses of gamma rays (0.2, 0.5, 1, 2, 4, 6, 8, and 10 Gy), and the m⁶A levels of *NCOA4* mRNAs were detected at 1, 3, 6, 12, 24, and 48 h after irradiation, while the m⁶A levels of *NCOA4* mRNAs in sham-exposed HUVECs were detected as a control

**Fig. 2 | The m⁶A modifications of *Ncoa4*, *Ate1* and *Fgf22* mRNAs as candidate biomarkers in response to irradiation in PBMCs from mice.** Schematic representation of m⁶A sites and correspondingly primer pairs designed to detect the m⁶A levels of *Ncoa4* (**a**), *Ate1* (**b**), and *Fgf22* (**c**) mRNAs in mice. Validation of primer specificity and measurement of RNA m⁶A levels of *Ncoa4* (**d**), *Ate1* (**e**), and *Fgf22* (**f**) mRNAs by MeRIP-qPCR assay in PBMCs of adult (6–8 weeks old) male C57BL/6 mice with 6.5 Gy TBI, at 1 day after gamma rays exposure, using the sham-exposed mice as control group (*n* = 5/group, 2 groups). Temporal responding curves of the relative m⁶A levels of *Ncoa4* (**g**), *Ate1* (**h**), and *Fgf22* (**i**) mRNAs in PBMCs from adult mice gamma-ray TBI model and the sham-exposed mice using *Ncoa4* primer 1, *Ate1* primer 2, and *Fgf22* primer 2 (*n* = 5/group, 33 groups). **j** The expression levels of *Ncoa4* mRNA in PBMCs from adult mice gamma-ray TBI model and sham-exposed mice using *Ncoa4* primer 1 (*n* = 5/group, 33 groups). **k** The linear relationship

between the relative products of SELECT (2^C_T values normalized to the 2^C_T value of 100% m⁶A) and m6A fraction served as a standard curve for the validation of m⁶A detection using SELECT method. **l** Measurement of the m⁶A levels of three highly confident m6A sites (A459, A761, and A781) at *Ncoa4* transcript in the PBMCs of adult (6–8 weeks old) male C57BL/6 mice at 7 and 14 days after 1, 2, 4 and 6.5 Gy gamma rays TBI, using the sham-exposed mice as control group (*n* = 5/group, 10 groups). The m⁶A fractions at A459 (left), A761 (center), and A781 (right) sites at *Ncoa4* mRNA in each group were calculated based on the corresponding standard curve. Data in **d**–**f** are presented as means ± SD and analyzed by two-sided Student's *t* test. For data in **g**–**l**, one-way ANOVA was applied, and adjusted by Dunnett's method. *P < 0.05, **P < 0.01, ***P < 0.001; n.s., not significant. Source data are provided.

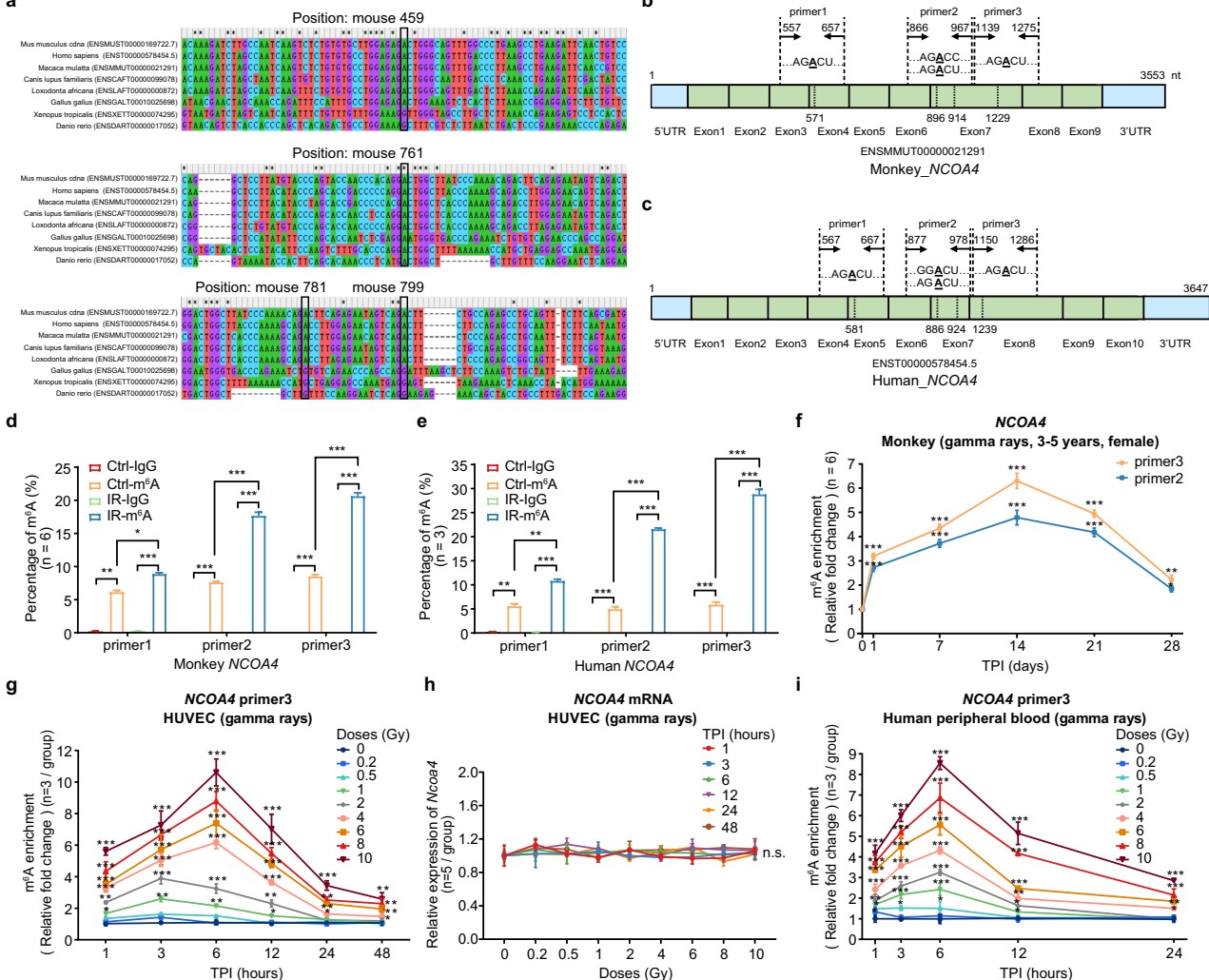

**Fig. 3 | Response of m⁶A-modified *NCOA4* mRNA in PBMCs from non-human primates (NHPs) and human cells exposed to radiation. a** Homology analysis of m⁶A methylated sites and conserved motifs around these sites in *Ncoa4*/*NCOA4* mRNA among different species. Schematic representation of m⁶A sites and correspondingly primers designed to investigate the m⁶A levels of *NCOA4* mRNA with exposure to radiation in NHPs (**b**) and human cells (**c**). **d** Validation of primers specificity and measurement of RNA m⁶A levels of *NCOA4* mRNA by MeRIP-qPCR in PBMCs from adult NHP gamma-ray TBI model (*n* = 6). **e** Validation of primers specificity and measurement of RNA m⁶A levels of *NCOA4* mRNA by MeRIP-qPCR in HUVECs exposed to 10 Gy gamma rays at 1 h post-irradiation, using unirradiated samples as control group (*n* = 3). **f** The relative m⁶A levels of *NCOA4* mRNAs in PBMCs from adult NHP gamma-ray TBI model exposed to 6.75 Gy gamma rays at 1, 7, 14, 21, and 28 days post-irradiation, using PBMCs from the unirradiated NHPs as

control group (*n* = 6). **g** Temporal responding curves of the relative m⁶A levels of *NCOA4* mRNA in HUVECs with various doses of gamma rays (0.2 to 10 Gy) and post-irradiation time (1, 3, 6, 12, 24, and 48 h) (*n* = 3/group, 49 groups). **h** The expression levels of *NCOA4* in HUVECs with various doses of gamma rays (0.2 to 10 Gy) exposure at 1, 3, 6, 12, 24, and 48 h and without irradiation (*n* = 3/group, 49 groups). **i** Temporal responding curves of the relative m⁶A levels of *NCOA4* mRNA in human peripheral blood cells with different doses of gamma rays (0.2 to 10 Gy) in vitro and post-irradiation time (1, 3, 6, 12, and 24 h) (*n* = 3/group, 41 groups). Data in **a** and **e** are presented as means ± SD and analyzed by two-sided Student's *t* test. MeRIP-qPCR assays in **f**–**i** were conducted using *NCOA4* primer 3. For data in **f** and **h**, one-way ANOVA was applied, and adjusted by Dunnett's method. *P < 0.05, **P < 0.01, ***P < 0.001. Source data are provided.

(Supplementary Data 3). The results by MeRIP-qPCR showed that the m⁶A curves of *NCOA4* mRNAs across different time points are elevated significantly after irradiation using either *NCOA4* primer 2 or primer 3, and the extend of elevation was clearly related to exposure dosage (Fig. 3g and Supplementary Fig. 6c). With exposure doses ≥4 Gy, the m⁶A levels of *NCOA4* peaked at 6 h post-irradiation; while with exposure doses of 1 or 2 Gy, the m⁶A levels of *NCOA4* peaked at 3 h post-irradiation (Fig. 3g and Supplementary Fig. 6c). When the radiation dose was less than 0.5 Gy, there was no significant difference in the m⁶A levels of *NCOA4* before and after irradiation. Particularly, dose-response analysis showed that the m⁶A levels of *NCOA4* mRNA increased 10-fold at 6 h after irradiation with a dose of 10 Gy (Supplementary Fig. 6d, e). Notably, we examined the expression levels of *NCOA4* mRNA in HUVECs after irradiation and found no significant differences in its expression at different radiation doses and post-irradiation times (Fig. 3h), thus excluding the interference of expression differences of *NCOA4* mRNA to m⁶A levels in MeRIP-qPCR assays.

We further assessed the dynamic change of the *NCOA4* m⁶A levels in an in vitro irradiation model of isolated peripheral blood cells from healthy human volunteers (*n* = 2). The peripheral blood cells were exposed to different doses of gamma rays (0.2, 0.5, 1, 2, 4, 6, 8, and 10 Gy), and total RNAs from which were collected at different time points post-irradiation (1, 3, 6, 12, and 24 h). Due to the sensitivity of blood cells to irradiation, the amount of total RNAs with treatment of different radiation doses and at multiple time points was first assessed and confirmed to be enough for MeRIP-qPCR analyses (Supplementary Fig. 6f). Consistent with the results in mice PBMCs and human HUVECs, the *NCOA4* m⁶A levels of peripheral blood cells after in vitro exposure to radiation were also significantly increased in a dose-dependent manner (Fig. 3i and Supplementary Fig. 6g; Supplementary Data 3). Together, these results demonstrate the excellent responding curves of RNA m⁶A levels of *NCOA4* in NHP PBMCs and human cells.

## Assessment of the responsive specificity of *Ncoa4* RNA m⁶A modification as an irradiation biomarker

We next sought to assess the responses of *Ncoa4* RNA m⁶A modification to the exposure of another type of ionizing rays—X-rays. Mice X-rays TBI models were constructed at different radiation doses (0.2, 0.5, 1, 2, 4, 6.5, and 10 Gy). Similar to gamma rays, X-rays exposure can also induce a significant increase of *Ncoa4* mRNA m⁶A levels, with a clear association with radiation dosage (Fig. 4a and Supplementary Fig. 7a). We also investigated the effect of X-rays exposure on the *NCOA4* m⁶A levels in human HUVECs. Similarly, the exposure to X-rays induced a significant increase, although slightly lower than the same doses of gamma rays, in the *NCOA4* m⁶A levels. The m⁶A levels of *NCOA4* were also significantly increased in a radiation dose-dependent manner in HUVECs (Supplementary Fig. 7h, i; Supplementary Data 3). Thus, these data suggest that two different types of ionizing rays, X-rays and gamma rays, may cause similar responses in the m⁶A modification of *Ncoa4* mRNA.

We then investigated the interference of other factors on the responding curves of m⁶A levels after irradiation to evaluate the reliability of m⁶A modification in *Ncoa4* mRNA as an irradiation biomarker under different scenarios. First, to assess the effects of immune system and inflammation on the responding curves of *Ncoa4* m6A modification, we constructed the immunodeficient nude mice gamma-ray TBI model and the inflammatory mice gamma-ray TBI model, respectively (see Methods). Similar to that in wild-type gamma-ray TBI mice, the *Ncoa4* m⁶A levels also showed a radiation dose-dependent increase in those immuno-compromised and inflammatory mice (Fig. 4b, c and Supplementary Fig. 7b, c). Second, to assess the effect of gender on the responding curves of RNA m6A modification, we constructed the female mice gamma-ray TBI model. Our results showed that the responding curves of m6A modification in female mice is similar with those in male mice (Fig. 4d and Supplementary Fig. 7d).

Third, to assess the effect of age on the responding curves of RNA m6A modification, we constructed the gamma-ray TBI model for young (~3 weeks) and old (~9 months) mice. In both young and old mice, the m⁶A levels of *Ncoa4* mRNA after irradiation showed the same responding curves as those in adult mice, although the elevation was relatively lower in young mice (Fig. 4e, f and Supplementary Fig. 7e, f). Based on the MeRIP-qPCR data in these mice TBI models, we found that the dose-response relationship of the m⁶A levels of *NCOA4* was not influenced by these confounding factors (Fig. 4g and Supplementary Fig. 7g).

Additionally, we examined the *NCOA4* m⁶A under other types of stress in HUVECs and found that the starvation, camptothecin (CPT), hypoxia and heat shock do not affect the m⁶A levels of *NCOA4* in HUVECs after acute radiation exposure (Supplementary Fig. 7j, m; Supplementary Data 3). Collectively, the response of *Ncoa4* m⁶A levels is robust to different types of ionizing rays, without major confounding effects due to differences in immune and inflammatory status, gender, age, or other types of stress.

## Response of *NCOA4* m⁶A modification in cancer patients undergoing radiotherapy

Next, we sought to investigate the dose response of *NCOA4* m⁶A in humans exposed to IR in vivo. To this end, a total of 33 cancer patients receiving local radiotherapy were recruited from the First Medical Center of Chinese PLA General Hospital (Beijing, China). According to the clinical requirements, the cancer patients received a single dose of 2 Gy each day and 5 times per week, eventually receiving a total of 50 Gy after treatment for 5 weeks in a row (Fig. 5a; Supplementary Data 4, 5). The PBMCs of patients were isolated at multiple time points before or during irradiation for subsequent experiments. Indeed, dose-response analyses by MeRIP qPCR assay using either the *NCOA4* primer 2 or primer 3 showed that the *NCOA4* m⁶A levels increases gradually with the accumulated radiation dose in these patients receiving local physical radiotherapy (Fig. 5b, c). Although radiotherapy was administered for up to 5 weeks, the m⁶A levels of *NCOA4* are elevated by about 5-fold when the fragmented radiation dose reached 50 Gy. Together, these results demonstrated the dose-response relationship of m⁶A modification of *NCOA4* in PBMCs from irradiated patients.

We also assessed the dynamics of *NCOA4* m⁶A modification in PBMCs from long-term low-dose exposed population. A collection of radiation workers (n = 6) with known long-term radiation exposure, and healthy volunteers (n = 16) without known irradiation exposure were recruited for this purpose (Supplementary Data 4; Methods). We observed that those radiation workers have higher *NCOA4* m⁶A levels than the healthy volunteers (Fig. 5d; Supplementary Data 4, 5). No significant confounding effect on *NCOA4* m⁶A levels was observed due to differences in age and gender (Fig. 5e, f). These findings therefore indicate the feasibility of *NCOA4* m⁶A modification in the screening of populations with long-term low-dose radiation exposure.

## Construction of radiation dose estimation models in mice and humans

We then sought to develop a practical biodosimetry assay that allows in vivo dose reconstruction from animals or humans exposed to unknown dose of irradiation. To this end, the m⁶A levels of *Ncoa4* in PBMCs after different doses of irradiation in vivo were used for developing algorithms by fitting in the experimental data points using the goodness of fit model. We first sought to construct the dose prediction models in gamma-ray TBI mice models. Because the m⁶A levels of *Ncoa4* were detected at different time points post radiation, we constructed the dose prediction models of absorbed doses for days 1, 3, 7, 14, and 28 separately using a two-order polynomial regression model (Supplementary Data 6). We found that these models achieved a high degree of consistency between the actual and estimated doses

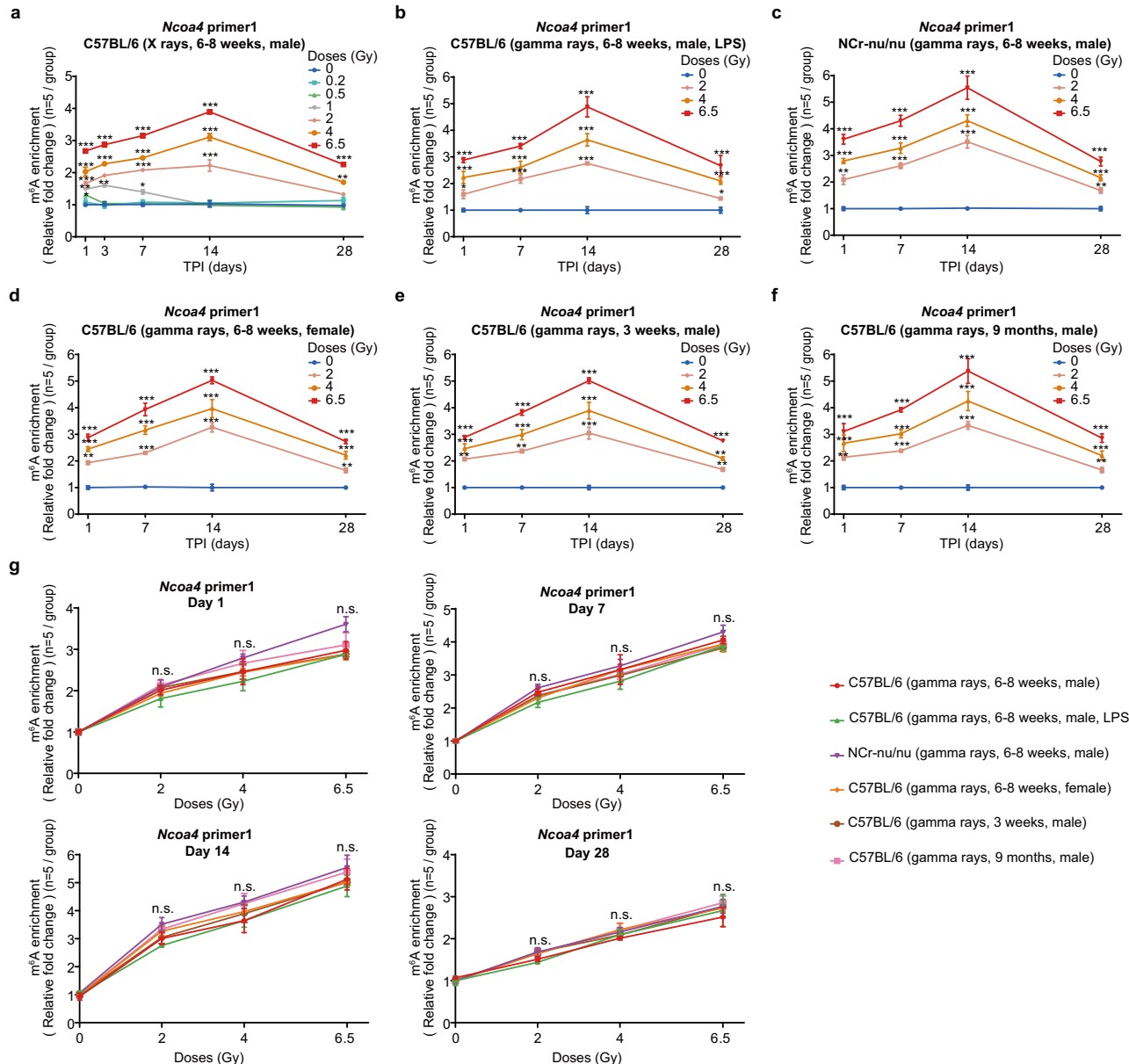

**Fig. 4 | Specificity assessment of m⁶A modification in *Ncoa4* mRNA as candidate biomarkers for irradiation exposure. a** Temporal responding curves of m⁶A modification of *Ncoa4* mRNA measured by MeRIP-qPCR assay in PBMCs from adult mice (6–8 weeks old) X-ray TBI model under different doses of irradiation exposure and sham-exposed mice ($n = 5$/group, 31 groups). PBMC, peripheral blood mononuclear cells; TBI, total-body irradiation. **b** Temporal responding curves of m⁶A modification of *Ncoa4* mRNA measured by MeRIP-qPCR assay in PBMCs from inflammatory adult mice gamma-ray TBI model and sham-exposed mice ($n = 5$/group, 13 groups). **c** Temporal responding curves of m⁶A modification of *Ncoa4* mRNA measured by MeRIP-qPCR assay in PBMCs from immunodeficient adult mice gamma-ray TBI model and sham-exposed mice ($n = 5$/group, 13 groups). **d** Temporal responding curves of m⁶A modification of *Ncoa4* mRNA measured by MeRIP-qPCR assay in PBMCs from female adult mice gamma-ray TBI model and sham-exposed mice ($n = 5$/group, 13 groups). Temporal responding curves of m⁶A

modification of *Ncoa4* mRNA measured by MeRIP-qPCR assay in PBMCs from younger (3 weeks) (**e**) and older (9 months) (**f**) mice gamma-ray TBI model and sham-exposed mice ($n = 5$/group, 25 groups). **g** The relative m⁶A levels of *Ncoa4* mRNA accompanying irradiation dose in different mouse models (referred to in **a**–**f**) at the same time points post-irradiation (days 1, 7, 14, and 28). Differential analyses were performed by comparing each group with the gamma-ray TBI model at each time point. MeRIP-qPCR assays were conducted by *Ncoa4* primer 1. The relative m⁶A levels of *Ncoa4* mRNA at the irradiated group were normalized to their corresponding sham-exposed control group. Data in (**a**–**g**) are presented as means ± standard deviation (SD) and was analyzed using one-way ANOVA, adjusted by Dunnett's method. *$P < 0.05$, **$P < 0.01$, ***$P < 0.001$, n.s., not significant. MeRIP-qPCR, methylated RNA immunoprecipitation in combination with real-time quantitative PCR. Source data are provided.

using either *Ncoa4* primer 1 ($R^2 = 0.919 - 0.966$) or primer 2 ($R^2 = 0.847 - 0.952$) (Fig. 6a and Supplementary Fig. 8a). Beside *Ncoa4*, prediction models built on the m⁶A levels of *Ate1* and *Fgf22* also showed high degree of consistency (Supplementary Fig. 8b–d). Considering we have monitored the m⁶A response at different time points post radiation, we further constructed an integrated prediction model that takes both m⁶A levels and time post irradiation (TPI) as input to

estimate the exposure doses using a binary three-order polynomial regression model. We found the integrated model constructed using either *Ncoa4* primer 1 or primer 2 nicely fit the observations, achieving $R^2$ of 0.932 (primer 1) or 0.924 (primer 2) (Fig. 6b and Supplementary Fig. 8e). This integrated model enables us to estimate the absorbed doses of mice which were exposed to irradiation as long as 28 days. Then, the receiver operating characteristic (ROC) analyses were used

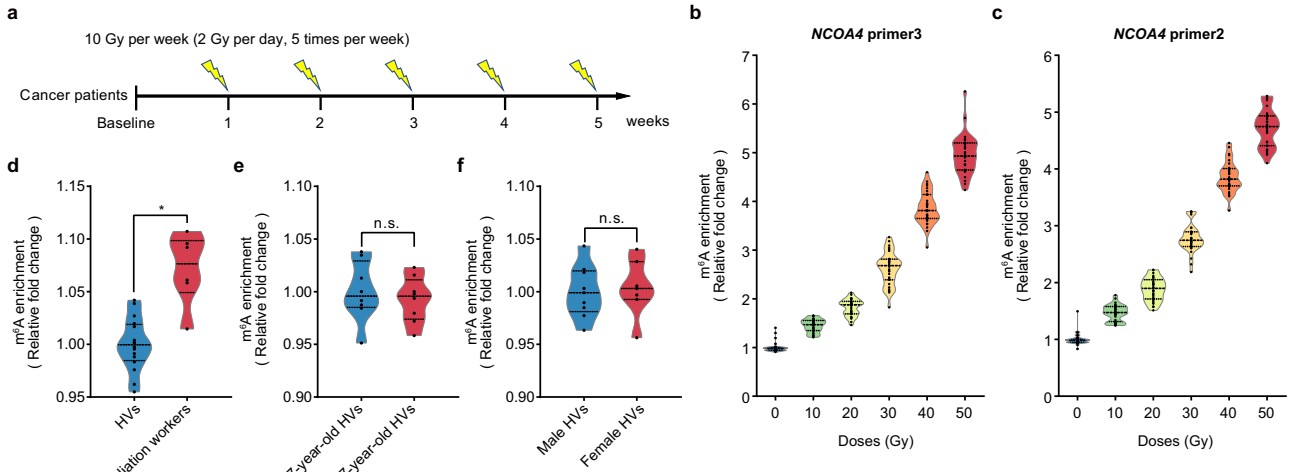

**Fig. 5 | Response of m⁶A-modified *NCOA4* mRNA in PBMCs from radiation-exposed humans. a** Schematic of clinical radiotherapy in cancer patients by partial body irradiation. Dose responding curves of the relative m⁶A levels of *NCOA4* mRNA in PBMCs from partial body irradiated cancer patients by *NCOA4* primer 3 (**b**) and primer 2 (**c**). Samples of control group (0 Gy) were collected within 1 week before the start of radiation (*i.e.*, baseline, $n = 30$). The relative m⁶A levels of *NCOA4* mRNA measured using *NCOA4* primer 3 in PBMCs from humans of different occupations (healthy volunteers [HVs, $n = 16$] *v.s.* radiation workers [$n = 6$]) (**d**), age (younger HVs [<27 years, $n = 8$] *v.s.* older HVs [≥ 27 years, $n = 8$]) (**e**), and gender (male HVs [$n = 9$] *v.s.* female HVs [$n = 7$]) (**f**). All data are presented as means ± standard deviation (SD) and analyzed by two-sided Student's *t* test. *$P < 0.05$, n.s., not significant. Source data are provided.

to assess the performance of the integrated model based on *Ncoa4* m⁶A levels on the prediction of absorbed irradiation dose. We first tested the model performances by fixing the TPI and found that the model could achieve very high accuracy with area under curve (AUC) varying from 0.992 to 1 with a dose cutoff of 4 Gy. We then removed the constrains on TPI and found the model can still achieve high accuracy within a dose cutoff range of 0.2 to 8 Gy (Fig. 6c, d). However, we found that a dose cutoff as low as 0.2 Gy is a challenge for this model, given that the corresponding AUC of 0.93 is slightly lower than other cutoffs (Fig. 6d).

Next, we sought to construct a dose estimation model in humans using the *NCOA4* m⁶A dataset in PBMCs from the cancer patients receiving fractionated irradiation therapy as we described in the previous session (Supplementary Data 7). We found that the correlation between the absorbed doses and m⁶A levels fits well with a two-order polynomial model, achieving AUCs of 0.901 and 0.877 using *NCOA4* primer 2 and primer 3 respectively (Fig. 6e, f). We further employed the ROC analysis to assess the performance of the model. By randomly repeating the five-fold cross-validation procedure for 100 times, we found the average AUCs by the two different sets of *NCOA4* primer pairs are both 0.970 (Fig. 6g, h). Taken together, these data demonstrate the feasibility of *NCOA4/Ncoa4* m⁶A modification in dose estimation in a dose range of ≥0.2 Gy both in gamma-ray TBI mice model and clinical patients receiving radiotherapy.

## Discussion

Fast and high-throughput biodosimeters are critical for the effective medical management following radiological accidents, where massive individuals could be exposed to irradiation with unknown doses. Here, we performed a transcriptome-wide screening for radiation-responding mRNA transcripts, whose m⁶A modification levels reveal significant changes after acute radiation exposure. We found that the RNA m⁶A levels of three genes, *Ncoa4*, *Ate1*, and *Fgf22*, show good dose reactivity, and therefore might serve as candidate biomarkers of radiation exposure. Especially, the m⁶A modification of *NCOR4* mRNA shows excellent performance in cross-species conservation, dose reactivity, responsive specificity, and long detectable duration in response to radiation. To our best knowledge, this is the first study

assessing the application of RNA m⁶A modification as candidate biomarkers in detecting radiation exposure.

Recently, several studies have revealed the close relationship between m⁶A RNA methylation and multiple genotoxic and non-genotoxic stresses. For examples, under the stress of UV, m⁶A RNAs and METTL16 (an m⁶A methyltransferase) are recruited to DNA damage sites and facilitate DNA repair by promoting nucleotide excision repair (NER) pathway[23,24]. Arsenite-induced oxidative stress can increase the expression of *WTAP* and *METTL14*, two genes encoding "writer" for RNA m⁶A, and the overall m⁶A levels, and then regulate the target RNAs in response to oxidative stress[35]. In response to ionizing radiation, METTL3 (another m⁶A methyltransferase) and FTO (an m⁶A demethylase) have been revealed to contribute to radiation resistance in an RNA m⁶A-mediated manner[36,37]. Besides, emerging evidence has demonstrated the crucial roles of RNA m⁶A upon other types of stresses, including hypoxia, therapeutic stress, metabolic stress and endoplasmic reticulum (ER) stress[38]. Therefore, these findings indicate the opportunity in focusing on RNA m⁶A modification to screen the candidate biomarkers in response to irradiation.

Indeed, a few pioneering studies have hinted the potential of RNA m⁶A modification as biomarkers of multiple diseases. With regard to cancer, an m⁶A score based on the expression of *IGF2BP2*, *IGF2BP3*, *KIAA1429*, *METTL3*, *EIF3H*, and *LRPPRC* was reported as an indicator of pancreatic tumor microenvironment status and was a potential biomarker for patients' prognosis[28]. The expression levels of *METTL3* were significantly elevated in the tissues of multiple cancers, and were associated with the patients poor outcomes[39]. Besides, evidences also suggested the expression profiles of m⁶A methylation-related genes to be a candidate biomarkers for metabolic abnormalities and cardiovascular diseases[40,41]. However, up to now, there is no study to directly evaluate the ability of RNA m⁶A modification itself as a biomarker. To our best knowledge, this is the first study assessing the application of RNA m⁶A methylation levels of specific genes as biomarkers, not limited to the radiation-related biomarkers.

As an epitranscriptomic modification, m⁶A RNA methylation exhibits a moderate level of temporal dynamics between epigenomic modification and gene transcription in response to stresses[42], which makes it a good biomarker of irradiation with early responding

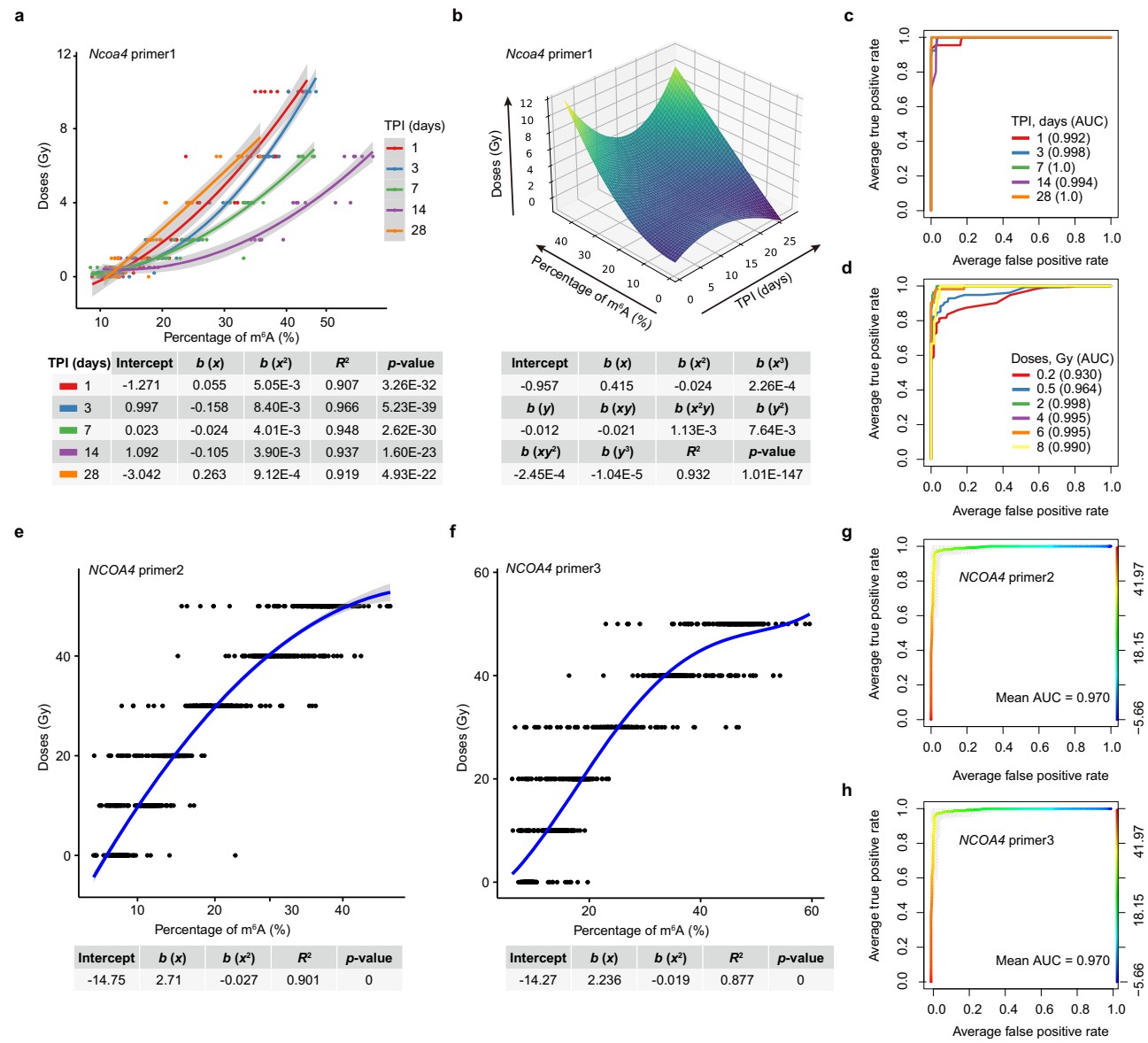

**Fig. 6 | Construction of radiation dose estimation models in mice and humans.**
**a** Fitting curves of two-order polynomial regression models for absorbed doses prediction in adult mice gamma-ray TBI model using the m[6]A percentages of *Ncoa4* mRNA, which were measured by MeRIP-qPCR assay using the *Ncoa4* primer 1. Actual irradiation doses were represented by scattered dots. X-axis: the observed m[6]A percentages of *Ncoa4* in PBMCs of mice; Y-axis: absorbed doses of irradiation. TBI, total-body irradiation; TPI, time post irradiation (days). Error bands represent 95% confidence interval (CI) estimated by the two-order polynomial regression model. **b** Fitting surface of a binary three-order polynomial regression model (integrated model) for absorbed doses prediction in adult mice gamma-ray TBI model taking both m[6]A percentage of mouse *Ncoa4* mRNA and TPI (days) as input. The m[6]A percentages of *Ncoa4* mRNA were measured by MeRIP-qPCR using the *Ncoa4* primer 1. X-axis: the observed m[6]A levels of *Ncoa4* in PBMCs of mice; Y-axis: the observed time post irradiation (days); Z-axis: absorbed doses of irradiation.

**c** Receiver operating characteristic (ROC) analysis of the performance of the integrated model on absorbed irradiation dose prediction in adult mice gamma-ray TBI model with fixed TPI (days). AUC, area under curve. **d** ROC analysis of the performance of the integrated model on absorbed irradiation dose prediction without TPI constrains in adult mice gamma-ray TBI model using the dose cutoff varying from 0.2 to 8 Gy. Fitting curves of two-order polynomial regression models for absorbed doses prediction in cancer patients receiving radiotherapy with multiple doses of X-ray exposure using *NCOA4* primer 2 (**e**) and primer 3 (**f**). Actual irradiation doses were represented by scattered dots. X-axis: the observed m[6]A levels of *NCOA4* in PBMCs of patients; Y-axis: the absorbed doses of irradiation. Error bands represent 95% CI estimated by the two-order polynomial regression model. (**g** and **h**) ROC analysis of the performance of the models on absorbed irradiation dose prediction in cancer patients receiving radiotherapy using *NCOA4* primer 2 (**g**) and primer 3 (**h**). Source data are provided.

timepoint and long detectable duration. In this study, we systemically assessed the dose-response relationships of the m[6]A methylation levels of *Ncoa4*, *Ate1*, and *Fgf22* mRNAs in PBMCs, and found that the m[6]A methylation levels of these three transcripts show different dynamic patterns after radiation exposure. Notably, the m[6]A levels of *Ncoa4* in mice increase continuously and reach to the highest value at day 14 after radiation, indicating its broader application scenarios with considerably longer time post-irradiation. We verified that the

response of m[6]A of *NCOA4* in monkeys and humans is close to its response in mice. Further, the well performance of the dose reconstruction model in the patients receiving radiotherapy provides assurance that m[6]A methylation of *NCOA4* can be applied to the cumulative dose estimation of fractioned radiation exposure over as long as 5 weeks. Notably, although this study focuses on acute radiation with medium and high dose exposure (>0.5 Gy), the good performance of *NCOA4* m[6]A modification in fractionated radiation

patients suggested its potential in low-dose radiation exposure over long time. Owing to the relatively early decreasing trend of m⁶A levels of *Ate1* and *Fgf22* mRNAs along with time, they might be suitable for the detection of early stage of acute radiation syndrome (ARS) which are capable of gauging absorbed radiation dose at a dose range of 1 to 6.5 Gy within 3 days post-irradiation.

The close relationship between the three m⁶A-based IR biomarker genes suggests possible mechanistic links between changes in their m⁶A levels and irradiation exposure. Indeed, we found all three genes, *NCOA4*, *ATE1* and *FGF22*, have interesting functional relevance with IR stress response, especially with DNA damage response. Specifically, NCOA4 (nuclear receptor coactivator 4) is a transcriptional coactivator of nuclear hormone receptors, which could inhibit the activation of DNA replication origins[43], prevent replication stress, maintain genome integrity, and reduce DNA damage[44]. ATE1 (arginyltransferase 1), a highly conserved gene across the eukaryotic domain, has been reported critical for suppressing the outcome of DNA mutagenesis during DNA-damaging stress[45,46]. FGF22 (fibroblast growth factor 22) has been reported to protect L02 cells from $H_2O_2$-induced oxidative damage via suppression of mitochondrial apoptosis pathways[47]. Together, the literature reviews of these genes indicate strong mechanistic connection between their altered m⁶A levels and irradiation exposure.

We observed significant differences in temporal response of m⁶A levels between mice exposed in vivo and samples irradiated ex vivo. This observation is consistent with previous studies on gene expression or other phenotypes of in vivo IR exposed mice or ex vivo irradiated samples[48–50]. One explanation for this phenomenon is that PBMCs in the in vivo exposed mice are continuously renewed by bone marrow and other emergency hematopoietic organs, such as spleen. Due to the fact that these hematopoietic organs are also irradiated in the TBI mice model, PBMCs in the circulatory system not only include cells directly exposed to irradiation, but also cells newly generated from the irradiated hematopoietic organs, which may be continuously under a ROS- and NOS-high microenvironment. This can lead to large differences in their molecular status responding to IR stress between in vivo exposed mice and ex vivo irradiated samples[51]. Besides, both the blood samples from whole body exposed mice and human cancer patients who received partial body fractionated radiation were used for the construction of dose estimation models. Although the cellular and molecular responses between the blood samples from total-body exposure and local exposure to IR is related, their relationship can be very complicated. Attempts have been made to mapping the exposure dose of local exposure to an estimated integral total-body exposure using γ-H2AX foci formation in PBMCs[52]. This study showed that the dosage conversion between total-body and local IR exposure varies remarkably among different body sites of local exposure. Another clinical study showed that although the deposited energy within the lung cancer patients is only half the energy measured within the rectal cancer patients, both cohorts have the same amount of in vivo chromosomal aberrations after one week[53]. The reason may be that the lungs hold a higher blood volume and blood flow than the pelvis, even though the lungs have a lower density. Therefore, unlike the total-body irradiation, the effect of local irradiation largely depends on the blood circulation in the irradiated area, and may lead to astonishing differences in DNA damage and chromosomal aberrations.

In this study, we mainly used MeRIP-qPCR assay to detect the levels of m⁶A RNA modification. To confirm the responding curves of *Ncoa4* m⁶A levels upon IR, we also employed a single-base resolution and non-antibody-based m⁶A mapping method SELECT. The results showed that three (A459, A761, and A781) of the four targeted sites exhibit significantly elevated m⁶A levels at both 7 and 14 days after irradiation, and the extend of elevation is clearly related to exposure dosage. Considering that the A761, A781 and A799 were located within a short region targeted by primer2 in the MeRIP-qPCR assay, the actual sites responding to IR are likely to be A761 and A781, but not A799. This

experiment not only confirmed the responding curves of *Ncoa4* m⁶A levels upon IR by MeRIP-qPCR assay, but also demonstrated the power of single-base m⁶A detection approaches in the m⁶A-based applications. Therefore, it is particularly necessary to develop more convenient, rapid and high-throughput m⁶A detection methods in the future. It is worthy to mention that a set of single-base and high-throughput m⁶A detection methods have been developed[54–56]. In addition, besides the m⁶A methylation, there are more than 170 types of RNA modification have been described thus far, including N1-methyladenosine (m¹A), N6-2′-O-dimethyladenosine (m⁶Am), 5-methylcytosine (m⁵C), pseudouridine (Ψ), and 5-hydroxymethylcytosine (hm⁵C)[57]. They are certainly also worthy of attention in the development of radiation-related biomarkers.

In summary, we systematically screened and evaluated the feasibility of RNA m⁶A modification in radiation dose assessment. In particular, the outstanding performance of m⁶A modification of *NCOA4* in dose reactivity, temporal dynamics, response specificity and cross-species conservation indicates its potential utility in radiation accident management and clinical application.

## Methods

### Human subjects

Three human cohorts were recruited in this study (Supplementary Data 4). The first is the acute radiation-exposure cohort, which consists of 33 cancer patients (including cervical, uterine, and vaginal cancers) who were scheduled to receive local radiotherapy (TOMO Therapy Hi·Art; Accuray, USA), and were recruited from the First Medical Center of Chinese PLA General Hospital (Beijing, China) between April and September in 2022. During local radiotherapy, the total dose received by these cancer patients gradually accumulated in the form of a single dose of 2 Gy (X-ray, 800 cGy/min; 5 times per week for 5 consecutive weeks). A total of 2 mL peripheral whole blood of elbow vein was collected within 1 week before the start of radiation at baseline (0 Gy) and thereafter at weeks 1, 2, 3, 4, and 5 (corresponding to 10, 20, 30, 40, and 50 Gy, respectively). The peripheral blood mononuclear cells (PBMCs) were isolated by density gradient centrifugation using the PBMC isolation kit (TBD sciences, China) according to the manufacturer's instructions. Total RNAs from PBMCs were extracted using the TRIzol reagent (Invitrogen, USA) for subsequent experiments.

The second is the non-radiation-exposed background cohort, which consists of 16 healthy volunteers without known irradiation exposure who were recruited between June and July in 2022 from the Beijing Institute of Radiation Medicine (Beijing, China). The third is the radiation-occupational-exposure cohort. A total of six operators of radioactive source, who involved in the operation of Cobalt-60 (⁶⁰Co) radioactive sources or engaged in radiotherapy occupations, were recruited in July 2022 from the Beijing Institute of Radiation Medicine (Beijing, China) and the Department of Radiation Oncology in the First Medical Center of Chinese PLA General Hospital (Beijing, China), respectively. A total of 2 mL peripheral whole blood of elbow vein was collected and PBMCs were isolated, and the total RNAs from PBMCs were extracted for subsequent experiments. In addition, we randomly selected two of the 16 healthy volunteers whose peripheral whole bloods were used in the irradiation exposure experiment in vitro. For details, please refer to the following section of "*Radiation exposure for human peripheral whole blood from healthy volunteers*".

This study was performed with the approval of the Medical Ethical Committee of Beijing Institute of Radiation Medicine (Beijing, China) and the Department of Radiation Oncology in the First Medical Center of Chinese PLA General Hospital (Beijing, China), and in accordance with the Council for International Organizations of Medical Sciences (CIOMS). All the patients receiving local radiotherapy, healthy volunteers and radiologic workers who participated in this study provided written informed consent, and their personal information on

demographic factors and clinical data were collected by structured questionnaire (Supplementary Data 4).

## Mice gamma-ray TBI models

For the construction of radiation exposure model in mice, the adult (6–8 weeks old) male C57BL/6 (Vital River Laboratories, China) were exposed to total body irradiation (TBI) with a $^{60}$Co gamma ray source at a dose rate of 69 cGy/min at Beijing Institute of Radiation Medicine (Beijing, China). For each radiation dose (0.2, 0.5, 1, 2, 4, and 6.5 Gy) and time point (days 1, 3, 7, 14, and 28), a minimum of five mice were used. Control mice were sham-exposed. The numbers of mice used in different radiation models were listed in Supplementary Data 2.

Specifically, in the stage I of m$^6$A profiling, a total of thirty C57BL/6 mice were divided into 6 groups, including a sham-exposed group and 5 groups with 6.5 Gy radiation at day 1, 3, 7, 14, and 28 after irradiation. In the stage II m$^6$A profiling, a total of fifteen C57BL/6 mice were divided into 3 groups, including a sham-exposed group and 2 groups with 2 Gy radiation at day 1 and 14 after irradiation.

In the validation stage of candidate m$^6$A sites using the methylated RNA immunoprecipitation in combination with real-time quantitative polymerase chain reaction (MeRIP-qPCR) assays, a total of 165 C57BL/6 mice were divided into 33 groups, including a sham-exposed group, 30 groups with different doses of gamma rays (0.2, 0.5, 1, 2, 4, and 6.5 Gy) at multiple time points (day 1, 3, 7, 14, and 28) after irradiation and 2 groups with 10 Gy gamma rays at 2 time points (day 1 and 3) after irradiation (since 10 Gy is the lethal dose for mice, fewer mice survived at day 7 after irradiation).

## Mice X-ray TBI models

To examine the response of RNA m6A modification to X-rays, the adult (6–8 weeks old) male C57BL/6 mice were exposed to TBI with an X-ray source at a dose rate of 119 cGy/min (X-Ray Irradiator; RAD Source, USA). A total of 155 C57BL/6 mice were divided into 31 groups, including a sham-exposed group and 30 groups with different doses of X-rays (0.2, 0.5, 1, 2, 4, and 6.5 Gy) at multiple time points (days 1, 3, 7, 14, and 28) after irradiation.

## Mice gamma-ray TBI models for evaluating different influencing factors

To examine the effect of age on m$^6$A levels, a total of 65 younger (3 weeks old) and 65 older (9 months old) male C57BL/6 mice were exposed to gamma-ray (TBI). These younger and older mice, respectively, were divided into 13 groups, including a sham-exposed group and 12 groups with different doses of gamma rays (2, 4, and 6.5 Gy) at multiple time points (day 1, 7, 14, and 28) after irradiation. To examine the effect of gender on m$^6$A levels, we constructed the female mice gamma-ray TBI model. A total of 65 adult (6–8 weeks old) female C57BL/6 mice were divided into 13 groups, including one sham-exposed group and 12 groups with different doses of gamma rays (2, 4, and 6.5 Gy) at multiple time points (days 1, 7, 14, and 28) after irradiation. To examine the effect of inflammatory conditions on m$^6$A levels, a total of 65 adult male C57BL/6 mice were treated (via intraperitoneal injection) with lipopolysaccharide (LPS, 5 mg/kg; Escherichia coli 055:B5, Sigma-Aldrich, USA) to mimic the state of acute inflammatory responses. Then, these treated mice were divided into 13 groups, including one sham-exposed group and 12 groups with different doses of gamma rays (2, 4, and 6.5 Gy) at multiple time points (days 1, 7, 14, and 28) after irradiation. We also constructed the immunodeficient mice TBI model for examining the effect of immune situation on m$^6$A levels. A total of 65 adult male NCr-nu/nu mice (Vital River Laboratories, China) were divided into 13 groups, including one sham-exposed group and 12 groups with different doses of gamma rays (2, 4, and 6.5 Gy) at multiple time points (days 1, 7, 14 and 28) after irradiation.

For each mice model, mice bloods were collected by retro-orbital plexus at each time point. PBMCs were then isolated by density gradient centrifugation using the PBMC isolation kit (TBD sciences, China) according to the manufacturer's instructions. Total RNAs from PBMCs were extracted using the TRIzol reagent (Invitrogen, USA) for subsequent experiments.

## NHP gamma-ray TBI model

To examine the response of RNA m$^6$A to irradiation in non-human primates (NHPs), we constructed the gamma-ray TBI monkey models. A total of six female rhesus monkeys (3–5 years old, $4.65 \pm 0.79$ kg body weight) (*Macaca mulatta*; SAFE Medical Technology, China) were anesthetized by intravenous injection of 3% pentobarbital sodium (1.0 mL/kg) and placed on their backs in wooden boxes, so that both sides of the monkey's body could receive 6.75 Gy $^{60}$Co gamma rays TBI at a dose rate of 63.98 cGy/min at Beijing Institute of Radiation Medicine (Beijing, China). Further, for homogenous dose distribution, the first half-dose was delivered by left-lateral exposure and the second half-dose was delivered by right-lateral exposure. A total of 2 mL peripheral whole blood of vein was collected before irradiation and at day 1, 7, 14, 21, and 28 post TBI. PBMCs were isolated by density gradient centrifugation using the PBMC isolation kit (TBD sciences, China) according to the manufacturer's instructions. Total RNAs from PBMCs were extracted using the TRIzol reagent (Invitrogen, USA) for subsequent experiments.

All animal experiments in this study were approved by the Animal Care and Use Committee of Beijing Institute of Radiation Medicine (Beijing, China). The acquisition, care, housing, use, and disposition of animals in research must comply with the applicable laws and regulations, institutional policies, and the international conventions to which China is a party.

## Hematological analyses

Approximately 20 μL of peripheral whole blood was collected through the tail vein for hematology analysis of mice without euthanasia. The white blood cell (WBC), lymphocyte (Lym), monocyte (Mono), red blood cell (RBC), and platelet (PLT) counts, as well as the hemoglobin (HGB) concentration were obtained using the automated URIT-5160Vet Hematology Analyzer (URIT Medical Electronic, China).

## Transcriptome-wide profiling of mRNAs and m$^6$A modifications in mice

Transcriptome-wide mRNAs expressions and m6A modification levels were quantified using the Arraystar Mouse RNA Epi-transcriptomic Microarray ($8 \times 60$ K, Arraystar, Rockville, MD, USA) based on the Arraystar's standard protocols. Briefly, the total RNAs were immunoprecipitated with anti-m$^6$A antibody (anti-m$^6$A rabbit polyclonal antibody, #202003; Synaptic Systems, Germany). The m$^6$A-modified RNAs eluted from the immunoprecipitated magnetic beads were set as the "IP". The un-modified RNAs recovered from the supernatant were set as "Sup". The RNAs in "IP" and "Sup" samples were then treated with RNase R, and labeled with Cy5 and Cy3, respectively, as RNAs in separate reactions using Arraystar Super RNA Labeling Kit. The RNAs were then hybridized onto the Arraystar Mouse RNA Epi-transcriptomic Microarray. After washing the slides, the arrays were scanned in two-color channels by an Agilent Scanner G2505C (Agilent Technologies, CA, USA).

Next, Agilent Feature Extraction software (version 11.0.1.1) was used to analyze the acquired array images. Raw intensities of "IP" (immunoprecipitated, Cy5-labelled) and "Sup" (supernatant, Cy3-labelled) were normalized with average of log2-scaled Spike-in RNA intensities. The "m$^6$A modification level" was then calculated as the percentage of modification based on the "IP" (Cy5-labelled) and "Sup" (Cy3-labelled) normalized intensities; whereas the "m$^6$A quantity" was calculated as the m$^6$A modification amount based on the "IP" (Cy5-

labelled) normalized intensities. The differentially m⁶A-methylated RNAs between two comparison groups were identified by filtering with the fold change (>2 or <0.5) and statistical significance ($P < 0.01$) thresholds. Finally, an unsupervised clustering analysis was performed by using *timeclust* function in the TCseq R package (v.1.14.0) to show the distinguishable m⁶A-modification pattern along different time points.

**Cells culture and treatments.** Human umbilical vein endothelial cells (HUVECs) were obtained from the China Center for Type Culture Collection (CCTCC; Wuhan City, China), and cultured in RMPI-1640 supplemented with 10% fetal bovine serum, 100 U/mL penicillin and 0.1 mg/mL streptomycin at 37 °C in a 5% $CO_2$ humidified atmosphere. To examine the dose and time relationship of RNA m⁶A modification to gamma ray, HUVECs were irradiated using a ⁶⁰Co gamma ray source at a dose rate of 69 cGy/min at Beijing Institute of Radiation Medicine (Beijing, China). HUVECs were divided into 49 group, including a sham-exposed group and 48 groups with different doses of gamma rays (0.2, 0.5, 1, 2, 4, 6, 8, and 10 Gy) at multiple time points (1, 3, 6, 12, 24, and 48 h) after irradiation. To examine the dose response of RNA m⁶A modification to X-rays, HUVECs were divided into 16 groups, including one sham-exposed group and 15 groups with different doses of X-rays (2, 6, and 10 Gy) at multiple time points (1, 3, 6, 12, and 24 h) after irradiation. To examine the effect of starvation on m⁶A levels, HUVECs were re-suspended in serum-free RMPI-1640 medium and seeded in 10-centimeter (cm) cell culture dish for 12 h. Then, the starved HUVECs were divided into 16 groups, including one sham-exposed group and 15 groups with different doses of gamma rays (2, 6, and 10 Gy) at multiple time points (1, 3, 6, 12, and 24 h) after irradiation while the starved HUVECs were still cultured in serum-free RMPI-1640 medium. To examine the effect of five different stress on m⁶A levels, HUVECs were treated with 1 μM camptothecin (CPT) (Sigma-Aldrich, USA) for 12 h to induce single-stranded DNA breaks and collected at multiple time point (1, 3, 6, 12, and 24 h). HUVECs were treated in an incubator at 42 °C for 1 h to induce a heat shock and collected at multiple time point (1, 3, 6, 12, and 24 h). HUVECs were treated in an incubator containing 5% $CO_2$, 1% $O_2$ and 94% $N_2$ for 24 h to induce a hypoxia stress and then followed by specified intervals (1, 3, 6, 12, and 24 h) of re-oxygenation in 5% $CO_2$ and 95% air incubator (21% $O_2$). A total of $1 \times 10^6$ HUVECs were needed in every group.

**Radiation exposure for human peripheral whole blood from healthy volunteers**

The peripheral whole blood of vein (~300 mL) collected from healthy volunteers was irradiated in 25 cm² cell culture flask at room temperature by a ⁶⁰Co gamma ray source at a dose rate of 69 cGy/min at Beijing Institute of Radiation Medicine (Beijing, China). The peripheral whole blood of vein was divided into 41 groups, including one sham-exposed group and 40 groups with different doses (0.2, 0.5, 1, 2, 4, 6, 8, and 10 Gy) of gamma ray radiation at multiple time points (day 1, 7, 14 and 28) after irradiation. Whole-blood from two healthy volunteers following irradiation in vitro were diluted with an equal volume of RMPI-1640 containing 10% fetal bovine serum in 25 cm² cell culture flask loosely capped and maintained on a 45-degree angle at 37 °C and collected at multiple time points (days 1, 7, 14, and 28) after irradiation.

**Evaluation of total RNA content from human peripheral whole blood**

Due to the sensitivity of blood cells to irradiation, we evaluated the content of total RNAs in blood cells treated with different radiation doses at multiple time points post-irradiation. There was no obvious difference in levels of total RNAs extracted from human PBMCs exposed to various radiation doses within 6 h post irradiation compared to in vitro un-irradiated human blood samples (Supplementary Fig. 6f). However, after 6 h post irradiation, the content of total RNAs in the blood showed a slightly drop with radiation dose higher than 4 Gy.

According to the volume of human blood sample, we calculated that the average total RNAs content of PBMCs obtained from 1 mL of human blood sample after irradiation was more than 1.5 μg, which was enough for quantifying human *NCOA4* mRNA by MeRIP-qPCR assays.

**MeRIP-qPCR assays**

The total RNAs were fragmented by sonication (Thermo Fisher Scientific, USA). Anti-m⁶A antibody (56593; Cell Signaling Technology, USA) or normal IgG (2729 s; Cell Signaling Technology, USA) was incubated with Protein A/G Beads (sc-2003; Santa Cruz Biotechnology, USA) at 4 °C for 4 h. After saving 500 ng of the total RNAs as input, the remaining RNAs were incubated with beads-antibody complex at 4 °C overnight in 500 μL of IP buffer (150 mM NaCl, 0.1% NP-40, 10 mM Tris, pH 7.4, 100 U RNase inhibitor) to obtain the m⁶A pull down portion (m⁶A IP portion) at a content of 2 μg in mice. Then, the m⁶A-modified RNAs were eluted with elution buffer (5 mM Tris-HCL pH 7.5, 1 mM EDTA pH 8.0, 0.05% SDS, 20 mg/mL Proteinase K). The m⁶A IP RNAs were then purified by Trizol and quantified using Nano-300 (Allsheng, China). The m⁶A IP RNAs and 500 ng input RNAs were used as templates in qRT-PCR assays, as described below. The IP enrichment ratio of a candidate m⁶A biomarker was calculated as the ratio of its amount in IP to that in the input generated from the same amount of RNAs. The relative level of candidate m⁶A biomarker at each exposure group was normalized to their corresponding un-irradiated control group. The qRT-PCR assays were conducted using the primers in Supplementary Data 1.

**Evaluation of primers for MeRIP-qPCR assays**

For the evaluation of primers in MeRIP-qPCR assays in mice, we designed two, three, and three primer pairs, respectively, for the detection of m⁶A modification levels of *Ncoa4* (ENSMUST00000169722.7), *Ate1* (ENSMUST00000094017.4), and *Fgf22* (ENSMUST00000020577.3), respectively. Among these primer pairs, two primer pairs targeting the m6A sites of *Ncoa4* mRNA (*i.e.*, primer 1 for A459; primer 2 for A761, A781 and A799), two primer pairs targeting the m6A sites of *Ate1* mRNA (*i.e.*, primer 2 for A767; primer 3 for A1782) and one primer pair targeting the m6A sites of *Fgf22* mRNA (*i.e.*, primer 2 for A370), respectively, showed clear signal of m⁶A enrichment and low IgG background. These primer pairs were therefore used for detection of m⁶A modification levels of these three genes, respectively.

For the evaluation of primers in MeRIP-qPCR assays in monkeys, we designed three primer pairs for the detection of m⁶A levels of *NCOA4* (ENSMMUT00000021291.4). Two primer pairs targeting the m6A sites of *NCOA4* mRNA (*i.e.*, primer 2 for A896 and A914; primer 3 for A1129) showed clear signal of m⁶A enrichment and low IgG background. These two primer pairs were therefore used for detection of m⁶A levels of *NCOA4* mRNA in monkeys.

For the evaluation of primers in MeRIP-qPCR assays in humans, we designed three, six, and one primer pairs, respectively, for the detection of m⁶A levels of *NCOA4* (ENST00000578454.5), *ATE1* (ENST00000224652.11), and *FGF22* (ENST00000215530.6), respectively. Among these primer pairs, two primer pairs targeting the m6A sites of *NCOA4* mRNA (*i.e.*, primer 2 for A886 and A924; primer 3 for A1239), three primer pairs targeting the m6A sites of *ATE1* mRNA (*i.e.*, primer 1 for A2015; primer 2 for A2015 and A2044; primer 5 for A4093) and one primer pair targeting the m6A sites of *FGF22* mRNA (*i.e.*, primer 1 for A1082), respectively, showed clear signal of m⁶A enrichment and low IgG background. These primer pairs were then used for detection of m⁶A levels of these three genes. All the primer pairs were listed in Supplementary Data 1.

**Real-time quantitative PCR assays**

Total RNAs were extracted using TRIzol™ Reagent and subjected to cDNA synthesis using MonScript™ RTIII All-in-One Mix kit (MR05101;

Monad, China). The qRT-PCR assays were performed using KAPA SYBR® FAST Universal kit (KK4601; KAPA Biosystems, USA) following the manufacturer's instructions. The relative RNA expression levels were normalized to *GAPDH*. The primers used in this study are listed in Supplementary Data 1.

### SELECT assays for single-base m⁶A detection

For the detection of m⁶A modification levels at the single-site level in *Nco*a4 mRNA transcripts, we designed four probe pairs targeting the A459, A761, A781 and A799 sites at *Nco*a4 transcript for SELECT assays. Total RNAs, RNA Oligo(A), or RNA Oligo(m⁶A) were mixed with 100 nM up probe, 100 nM down probe and 2 µL dNTP in 17 µL 1 × Reaction buffer (R202106M-03; Epibiotek, China). The RNA and probes were annealed by incubating mixture at a temperature gradient: 90 °C for 1 min, 80 °C for 1 min, 70 °C for 1 min, 60 °C for 1 min, 50 °C for 1 min, and then 40 °C for 6 min. Subsequently, a 3 µL of mixture containing 0.3 µL SELECT DNA polymerase, 0.47 µL SELECT™ ligase and 10 nmol ATP was added in the former annealed mixture to the final volume of 20 µL. The final reaction mixture was reacted at 40 °C for 20 min, and then was denatured at 80 °C for 20 min and kept at 4 °C. A series of standard RNA mixture with known m⁶A fraction by mixing RNA Oligo(A) with RNA Oligo(m⁶A) and total RNAs were used as templates in qRT-PCR assays using the Select primers, as described below. The m⁶A fraction at the single-site level in *Nco*a4 mRNA was calculated by the standard curve (Supplementary Fig. 4a–d).

### Immunofluorescence assays

After irradiation, PBMCs from mice were collected at the indicated times, washed twice in PBS, fixed with 4% paraformaldehyde for 10 min, permeabilized with 0.5% Triton X-100-PBS for 5 min, and then incubated with primary rabbit polyclonal anti-Phospho-Histone H2AX (Ser139) (1:400, 100-384; NOVUS, USA) for 20 min at 25 °C. Staining was conducted with anti-rabbit secondary antibodies conjugated to Rhodamine (Jackson ImmunoResearch, USA) for 20 min at 25 °C. Nuclear counterstaining was conducted with DAPI, and digital images were obtained using a fluorescence microscope (Nikon, Japanese) and fluorescence intensity was analyzed by imageJ.

### Western blotting assays

Proteins from PBMCs in mice were extracted using cell lysis buffer (Cell Signaling Technology, USA) containing protease inhibitor cocktail (Roche, Switzerland). Protein samples were resuspended in Laemmli buffer (63 mM Tris-HCl, 10% glycerol, 2% SDS, 0.0025% bromophenolblue, pH 6.8) and electrophoresed on SDS-polyacrylamide gels. Then, proteins were transferred to polyvinylidene fluoride (PVDF) membranes (Millipore, USA). After that, the membranes were blocked with 5% nonfat milk (Difco, USA) in TBST at 25 °C for 1 h. Primary antibodies (1:1000, 100-384; NOVUS, USA) were incubated at 4 °C overnight. Anti-rabbit secondary antibodies (1:2500, A21020; Abbkine, China) conjugated to horseradish peroxidase (HRP) were incubated at 25 °C for 1 h. The immunoreactive bands were detected using Super-Signal™ West Pico chemiluminescent substrate kit (Thermo Fisher Scientific, USA) and Western blotting detection system (Tanon 5200, China).

### Model fitting for radiation dose estimation in mice and humans

Polynomial regression models were used for the estimation of radiation doses. To construct radiation doses estimation models in a given day post irradiation both in mice and humans, we employed a two-order polynomial regression model to fit the actual dose of exposure with the observed m⁶A levels of *NCOA4* in PBMCs. To construct an integrated model that takes both m⁶A levels and time post irradiation as input for the doses estimation in mice, a binary three-order polynomial regression model was introduced. The fitness between the

actual and estimated doses were evaluated using the coefficient of determination ($R^2$). Besides, 100 times repeated five-fold cross-validation procedure and receiver operating characteristic (ROC) analyses were used to assess the performance of the estimation model on the prediction of absorbed irradiation dose. The two-order polynomial regression models were implemented using R language (v4.0.3), the binary three-order polynomial regression model was implemented using the "*NumPy*" package in Python (v3.8), and the ROC analysis was performed using the "*ROCR*" R package.

### Statistical analyses

Statistical analyses were performed by Prism 8.0 software (GraphPad, USA). The results are shown as the means ± standard deviation (SD) of at least three biological replicates. Comparisons between two groups were analyzed by unpaired Student's $t$ tests. One-way analysis of variance (ANOVA) followed by Dunnett's test was used for comparisons among multiple groups. A $P$ value less than 0.05 was considered to be statistically significant for all the tests.

### Reporting summary

Further information on research design is available in the Nature Portfolio Reporting Summary linked to this article.

### Data availability

The raw and processed mRNA expression and m⁶A modification datasets generated by Arraystar Mouse RNA Epi-transcriptomic Microarray in stages I and II mice TBI experiments have been deposited in the GEO database under accession code GSE225404 and GSE225405. The MeRIP-PCR and SELECT data of mice, monkey and human generated in this study are provided in the Supplementary Data file. Source data are provided with this paper.

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

## Acknowledgements

We thank all the patients and volunteers for donating blood samples. We sincerely thank staff at the Institute of Animal Care and Use Committee of Beijing Institute of Radiation Medicine for animal care and staff at the Beijing Institute of Radiation Medicine for irradiation management and operation. This work was supported by grants from the Natural Science Foundation of Beijing (5202025), the General Program (32270714 and 32200511) of the Natural Science Foundation of China (https://www.nsfc.gov.cn), and grants from Beijing Institute of Radiation Medicine (BWS21J022, AWS21J003 and AMMS-QNZD-2022-002).

## Author contributions

G.Q.Z. was the principal investigator who conceived the study and obtained financial supports. G.Q.Z., Y.M.L., H.X.C., X.Z., and W.Y. designed the study. Y.M.L., S.A.J., H.L., C.Q., and Y.F.L. were responsible for bioinformatics analyses of epi-transcriptomic microarray data. H.X.C., X.Z., Q.Z., Z.Y.Y., X. S, and H.H.H constructed the mice TBI models. H.X.C., X.Z., and Q.Z. performed the MeRIP-based m6A detection experiments in mice, monkey, and human blood samples. X.Z., Q.Z., and R.J.H. performed the m6A detection-related experiments in HUVECs. H.X.C., Y.M.L., X.Z., C.J.F., W.Y., and Q.Q.W. performed the statistical analyses, interpreted the results, and drafted the manuscript. G.Q.Z. approved the final version of the manuscript.

## Competing interests

The authors declare no conmpeting interests.
