## [Peer Review File · Nature Communications]

REVIEWER COMMENTS

Reviewer #1 (Remarks to the Author):

Chen et al, identified radiation dose dependent increase in RNA N-6 methyl adenosine (N6A) in selected genes and demonstrated the potential to develop as a biomarker providing readout of radiation response antibody and sequencing of enriched hypo and hyper adenosine methylated genes in peripheral blood in mouse, led to the identification of changes in N6A methylation at specific positions in three candidate genes, Ncoa4, Ate1, and Fgf22, which was followed up for their dose and time dependent changes. RNA isolated from whole blood collected from mice exposed to total body irradiation at dose range and time points relevant to triage and clinical decision making in humans after a major radiological event was used for discovery. Authors focused on site specific modifications in three candidates, including Noca1 gene in blood collected from irradiated non-human primates and human cancer patients who received therapeutic radiation. The robustness of the response was validated with various confounders, such as age, gender and acute inflammatory responses. The data obtained from mice blood samples were used to develop a dose reconstruction algorithm and compared that with the response noted at corresponding sites in blood collected human cancer patients received partial body fractionated radiation.

Overall, the study is comprehensive, and the data looks original, and the primary data file is attached. The dose dependent increase in N6A is sustained until 14 days after exposure to ionizing radiation. This is a strength considering several of the biomarkers that are under development show response only in the first week (messenger RNA expression sustain to 2-3 days and miRNA-based markers to 7 days). Another strength is that the assay is internally controlled, using unmodified adenosine at the same site as control, hence permit normalization without baseline sampling. The drawbacks include that the dose response at lower dose range is rather small, and the assay is technically involved, and it needs at least a day to get the result. Thus, the utility of the assay for triage or clinical decision making after a large-scale nuclear event is still not evident. Nonetheless, the findings have some scientific merit, experiments are well controlled, and the response has been validated in multiple models, including specimens from human cancer patients and radiation workers.

Major comments:

1. Figure 5: The data generated from irradiated human blood specimens are missing (no figure 5d, although there is a reference in figure legend). Also, need to include specifics of the samples collected from radiation workers (physical dosimetry parameters, demography etc.).
2. Supp Fig. 4: Gamma-H2AX signals were used as reference dosimetry assay. How many cells survived 7, 14 and 28 days after 6.5 Gy in vitro exposure to sample? Figure legend says 0.2 -10 Gy as dose, but the data is shown for up to 6.5 Gy only. Gamma-H2AX does indeed show dose response, however the

signals are transient (peak at 30 min and lost after 24 hours). Since the kinetics in mice is followed for days, lymphocyte depletion kinetics will serve as better reference.

3. There is large difference in temporal response after in vivo exposed mice versus ex vivo irradiated samples. The N6A at specific sites in Ncoa4 in blood from the irradiated mice peaked at 14 days, however that in cell lines (HUVEC) and PBMCs peaked at 6 hours. Please explain.

Minor comments

4. The blood samples after whole body exposed mice was used to evaluate the dose response in human cancer patients who received partial body fractionated radiation. Is the response from partial body exposure is comparable with that of whole-body exposure? Maybe worth mentioning that radiation was delivered at a conventional dose rate and the circulating blood might get exposed to radiation over the time.

5. Primary data file has been attached. It would be helpful to add the time points as well in the excel sheet (e.g., m6A enrichment (IR/0 Gy) in the table. Was the baseline (0 Gy) collected for each time points were used for normalization?

6. Fig 3 f: Please provide radiation dose at which PBMC collected from NHPs are irradiated in the figure and/or figure legend (in addition to that in materials and method section).

7. It would be helpful to discuss the functional relevance of the N6A methylation in those specific candidate genes and sites. Are these functional biomarkers of radiation dose response or are these responses are simply correlative? Is there any mechanistic link in the context of radiation exposure, other than generic stress response?

Reviewer #2 (Remarks to the Author):

The manuscript describes discovery of the N6-methyladenosine modification-based biomarkers for the estimation of the dose of absorbed ionizing radiation. The authors perform a genome-wide screen of radiation-responding mRNA transcripts and show that the statuses of m6A methylation of several genes (Ncoa4, Ate1 and Fgf22) in murine blood mononuclear cells (PBMCs) exhibit strong dose-response relationship after acute radiation exposure. The author's subsequent analysis demonstrates that the level of m6A methylation of NOCA4 (human orthologue of Ncoa4) may serve as a reliable biomarker for estimation of the dose of ionizing radiation in patients receiving radiation therapy. As far as I can ascertain, the study is generally technically sound, novel and interesting. Conceptually, the feasibility of using m6A and other RNA modifications for radiation accident management and radiotherapy

represents potentially very important advance for medicine. The manuscript is also clearly written and the results are nicely presented.

Despite, overall, I do feel that this is an important and timely study; I have two main issues with this manuscript:

1. In my opinion, the authors should confirm their main results with an orthogonal (preferably single-base resolution and not-antibody-based) method of m6A mapping on, at least the Ncoa4/NOCA4 transcript. As several such techniques have been published during the recent years (e.g. Nat Methods 19, 1590–1598 (2022); Nat Biotechnol 40, 1210–1219 (2022); Nat Methods 12, 767–772 (2015)), this seems feasible.

2. The paper is purely descriptive and the authors do not show any attempt to explain potential mechanisms for the phenomenon they observe (the correlation of m6A methylation of the identified transcripts with the dose of ionizing radiation). Although, I feel that, since potential use of m6A modification as a biomarker for the radiation dose may be interesting by itself, such descriptiveness is not necessarily precludes the publication of this work in a high impact journal, the authors should, at least, speculate on potential biological meaning of their results and, ideally, suggest a hypothetical model explaining their observations. Therefore, the Discussin should be expanded and rewritten correspondingly, in my opinion.

Point-by-Point Responses

We'd like to thank the editor and both reviewers for their valuable comments. We greatly appreciate the input, and have enclosed a revised version that would hopefully address the concerns in the reviews. Especially, we have conducted a single-base resolution and non-antibody-based m⁶A mapping method SELECT, further confirming the responding curves of *Ncoa4* m⁶A methylation levels upon IR. We have also added necessary information about the radiation workers and sample collection and rephrased the unclear parts in the manuscript. Finally, we discussed the possible mechanistic roles of *Ncoa4*, *Ate1* and *Fgf22* in the cellular response to IR stress. We detail our responses to the reviews below.

Many thanks to Reviewer #1 for the detailed review of this manuscript. In response to your specific comments:

Major comments:

1. "Figure 5: The data generated from irradiated human blood specimens are missing (no figure 5d, although there is a reference in figure legend). Also, need to include specifics of the samples collected from radiation workers (physical dosimetry parameters, demography etc.)."

Response: Thank you for the suggestion. In the manuscript, Fig. 5d. is placed at the lower left of Fig. 5, and the data used for generating Fig. 5d is put in the 'Data file' (sheet 'Fig. 5, d to f'), which contains the m⁶A enrichment levels (relative fold change) of *NCOA4* mRNA (primer 3) in PBMCs from healthy volunteers and radiation workers. Besides, the raw data and intermediate results during calculation were also placed at the bottom of the sheet. As suggested by the Reviewer, we have added the physical dosimetry parameters and demography information of the samples collected from radiation workers to Supplementary Data 4 (B) in the revised manuscript.

2. "Supp Fig. 4: Gamma-H2AX signals were used as reference dosimetry assay. How

many cells survived 7, 14 and 28 days after 6.5 Gy *in vitro* exposure to sample? Figure legend says 0.2-10 Gy as dose, but the data is shown for up to 6.5 Gy only. Gamma-H2AX does indeed show dose response, however the signals are transient (peak at 30 min and lost after 24 hours). Since the kinetics in mice is followed for days, lymphocyte depletion kinetics will serve as better reference.”

Response: We apologize for the ambiguous description about the dose range of IR exposure used for gamma-H2AX experiments in the original manuscript. Because almost all mice died shortly (≤ 3 days) after 10 Gy TBI, we discarded this dose in the following gamma-H2AX measurement. We have accordingly modified the legend of Supplementary Fig. 5a to “Immunofluorescence staining of gamma-H2AX (γ H2AX) in PBMCs from adult mice gamma-ray TBI model with a broad dose range of gamma rays (0.2 to **6.5 Gy**) was evaluated at day 1, 3, 7, 14 and 28 after irradiation (n = 5)” in the revised manuscript. It’s really an excellent suggestion of using lymphocyte depletion kinetics as a reference of the temporal changes of RNA m⁶A levels after IR exposure. Actually, we have performed hematological analyses to measure the temporal changes of multiple blood cells (including lymphocytes, monocytes, red blood cells, platelet, etc.) in Supplementary Fig. 1. We found that the number of lymphocytes is markedly reduced at the first day post irradiation in a dose-dependent manner (Supplementary Fig. 1a-c), and the reduction lasted for at least 14 days after irradiation for mice exposed to a dose ≥ 2 Gy. These lymphocyte depletion kinetics after IR exposure are consistent with previous studies (Avicenna J Med Biotech., 2013, PMID:23919119; Sci Transl Med., 2020, PMID:32669422) and can be used as reference for the temporal responses of RNA m⁶A levels.

3. “There is large difference in temporal response after *in vivo* exposed mice versus *ex vivo* irradiated samples. The N6A at specific sites in Ncoa4 in blood from the irradiated mice peaked at 14 days, however that in cell lines (HUVEC) and PBMCs peaked at 6 hours. Please explain.”

Response: This is a very interesting point. Indeed, we observed that samples from *in vivo* exposed mice and *ex vivo* irradiated samples showed markedly different

dynamics of m⁶A methylation. This observation is consistent with previous studies on gene expression or other phenotypes of *in vivo* IR exposed mice or *ex vivo* irradiated samples. For example, it has been reported that the γ -H2AX response is maximal 30 minutes after exposure and declines over a period of hours as the cells repair the damage. One study has obtained a linear response proportional to the initial radiation dose 48 hours after exposure in peripheral blood lymphocytes irradiated *ex vivo* (Adv Space Res., 2009, PMID:20046946). Another study has reported that γ -H2AX foci in murine skin may be a useful biodosimeter to determine dose at times up to 7 days after radiation exposure *in vivo* (Radiat Res., 2010, PMID:20041754). Also, in a rhesus macaque (*Macaca mulatta*) model, γ -H2AX foci were proportional to initial irradiation doses and statistically significant responses were observed until 14 days after 8.5 Gy in lymphocytes and until 9 days after 8.5 Gy in plucked hairs (PLoS One., 2010, PMID:21124906). One explanation for this phenomenon is that PBMCs in the *in vivo* exposed mice are continuously renewed by bone marrow and other emergency hematopoietic organs, such as spleen. These hematopoietic tissues were also irradiated in the TBI mice model. Therefore, PBMCs in the circulation system comprise not only the cells that were directly exposed to irradiation, but also cells newly generated from the irradiated hematopoietic tissues, which are likely to be continuously under a ROS- and NOS-high microenvironment. This can lead to large differences in their molecular status responding to IR stress between *in vivo* exposed mice and *ex vivo* irradiated samples (Radiat Oncol., 2019, PMID: 31892333). We have added this point in the Discussion section (page 26; lines 527-538) in the revised version.

Minor comments:

4. “The blood samples after whole body exposed mice was used to evaluate the dose response in human cancer patients who received partial body fractionated radiation. Is the response from partial body exposure is comparable with that of whole-body exposure? Maybe worth mentioning that radiation was delivered at a conventional dose rate and the circulating blood might get exposed to radiation over the time.”

Response: Also a very interesting question. Indeed, the cellular and molecular responses between the blood samples from total-body exposure and local exposure to IR is related. However, the relationship might be very complicated. Attempts have been made to mapping the exposure dose of local exposure to an estimated integral total-body exposure using γ -H2AX foci formation in PBMCs (Int J Radiat Biol., 2007, PMID: 17729159). This study showed that the dosage conversion between total-body and local IR exposure varies remarkably among different body sites of local exposure. Another clinical study showed that although the deposited energy within the lung cancer patients was only half the energy measured within the rectal cancer patients, both cohorts had the same amount of *in vivo* chromosomal aberrations after one week (Radiat Prot Dosimetry., 2011, PMID: 21131662). As suggested by the reviewer, the reason may be that the lungs hold a higher blood volume and blood flow than the pelvis, even though the lungs have a lower density. As a result, different from total-body irradiation, local irradiation depends greatly on the blood circulation in the irradiated area and can lead to astonishing differences in DNA damage and chromosomal aberrations. In conclusion, the relationship between the cellular and molecular responses of the blood samples from total-body exposure and local exposure to IR can be very complicated, the irradiation site dependence has to be considered. We have added text to the Discussion section about the relationship between dose estimation of total-body and local irradiation (pages 26-27; lines 538-554).

5. “Primary data file has been attached. It would be helpful to add the time points as well in the excel sheet (e.g., m⁶A enrichment (IR/0 Gy) in the table. Was the baseline (0 Gy) collected for each time points were used for normalization?”

Response: We apologize for the ambiguous description in the original manuscript. We did use the m⁶A enrichment levels of samples with 0 Gy as baselines to normalize the m⁶A levels of sample with > 0 Gy IR at the same time point. In the original manuscript, by phrasing “IR/0 Gy”, we actually tried to express the meaning of “IR samples versus 0 Gy samples”. To avoid confusion, we have now changed “IR/0 Gy”

in all related table headers to “normalized to baseline (0 Gy)” in the revised manuscript (Data file Fig. 2g-i; Fig. 3f-g; Fig. 3i; Fig. 4a-f; Fig. 5b-c; Fig. S3, d-e; Fig. S6c; Fig. S6g; Fig. S7, a-f; Fig. S7, h-m.).

6. “Fig 3 f: Please provide radiation dose at which PBMC collected from NHPs are irradiated in the figure and/or figure legend (in addition to that in materials and method section).”

Response: Thank you for the reminder. The radiation dose for NHPs is 6.75 Gy. We have added the following text to the figure legend of Fig. 3f in the revised manuscript (page 56, lines: 1136-1138):

>>> The relative m⁶A levels of *NCOA4* mRNAs accompanied with prolonged post-irradiation time in PBMCs from adult NHP gamma-ray TBI model exposed to 6.75 Gy gamma rays at 1, 7, 14, 21, and 28 days post-irradiation, using PBMC samples from unirradiated NHPs as control group (n = 6).

7. “It would be helpful to discuss the functional relevance of the N⁶A methylation in those specific candidate genes and sites. Are these functional biomarkers of radiation dose response or are these responses are simply correlative? Is there any mechanistic link in the context of radiation exposure, other than generic stress response?”

Response: Thank you for the excellent suggestion. Based on your suggestion, we have conducted thorough literature survey on the known functionalities of the candidate genes that might have mechanistic links with IR stress response. Indeed, we found all three genes, *NCOA4*, *ATE1* and *FGF22*, have interesting functional relevance with IR stress response, especially with DNA damage response.

Specifically, *NCOA4* (Nuclear Receptor Coactivator 4) is a transcriptional coactivator of nuclear hormone receptors, which could inhibit the activation of DNA replication origins (Mol Cell., 2014, PMID:24910095) and prevent replication stress, maintain genome integrity, and reduce DNA damage (Cell Rep., 2022, PMID: 35977492).

ATE1 (arginyltransferase 1), a highly conserved gene across the eukaryotic domain, has been reported critical for suppressing the outcome of DNA mutagenesis during

DNA-damaging stress (e.g., Nat Commun., 2023, PMID:36709327; Cell Death Dis., 2016, PMID:27685622). FGF22 (fibroblast growth factor 22) has also been reported to protect L02 cells from H₂O₂-induced oxidative damage via suppression of mitochondrial apoptosis pathways (Protein Expr Purif., 2018, PMID:29627393). Taken together, the literature analyses on these genes suggest that there are strong mechanistic links between their altered m⁶A methylation levels and irradiation exposure. We have now added this content to the Discussion section of the revised manuscript (pages 25-26, lines 513-526).

Reviewer #2.

Many thanks to Reviewer #2 for your detailed review of this manuscript. We've addressed these issues as follows:

1. "In my opinion, the authors should confirm their main results with an orthogonal (preferably single-base resolution and not-antibody-based) method of m⁶A mapping on, at least the *Ncoa4*/NOCA4 transcript. As several such techniques have been published during the recent years (e.g. *Nat Methods* 19, 1590–1598 (2022); *Nat Biotechnol* 40, 1210–1219 (2022); *Nat Methods* 12, 767–772 (2015)), this seems feasible."

Response: Thank you for the excellent suggestion. Indeed, the MeRIP-qPCR method we used in the manuscript could not locate the specific m⁶A sites being hyper-methylated upon IR. Although this does not affect the validity of using targeted region(s) for m⁶A detection and their application as IR biomarkers, a single-base resolution m⁶A detection method will be very helpful for targeting the specific m⁶A sites of actual effect. As suggested by the reviewer, we have conducted a single-base resolution and non-antibody-based m⁶A mapping method SELECT (*Angew Chem Int Ed Engl.*, 2018, PMID:30345651) to further confirm the responding curves of *Ncoa4* m⁶A methylation levels upon IR. Specifically, the up-stream and down-stream probes, RNA oligo (A) and RNA oligo (m⁶A) were designed respectively according to the four highly confident m⁶A modification sites (A459, A761, A781 and A799) at *Ncoa4* transcript. Then, SELECT qPCR assay were conducted to measure the m⁶A levels of the four specific m⁶A modification sites at *Ncoa4* mRNAs in PBMCs from mice at 7 and 14 days after treatment with varying doses (1, 2, 4 and 6.5 Gy) of gamma rays TBI. The results showed that three of the four targeted sites exhibited significantly elevated m⁶A levels both at 7 and 14 days after irradiation, and the extend of elevation was clearly related to exposure dosage (Fig. S4). Considering the A761, A781 and A799 were located within a short region targeted by primer2 in the MeRIP-qPCR assay, the actual sites responding to IR are likely to be A761 and A781, but not A799. We believe that this experiment not only confirms our previous results on the responding curves of *Ncoa4* m⁶A methylation levels upon IR, but also demonstrates

the power of single-base m⁶A detection approaches in the m⁶A-based applications. We have added the related texts to the corresponding Results (page 13, lines 258-267) and Methods sections (page 40, lines 828-843), respectively, in the revised version. Besides, we also discussed the power of single-base m⁶A detection methods in the future researches and applications of m⁶A in the Discussion section (pages 27-28, lines 556-566).

2. “The paper is purely descriptive and the authors do not show any attempt to explain potential mechanisms for the phenomenon they observe (the correlation of m⁶A methylation of the identified transcripts with the dose of ionizing radiation).

Although, I feel that, since potential use of m⁶A modification as a biomarker for the radiation dose may be interesting by itself, such descriptiveness is not necessarily precludes the publication of this work in a high impact journal, the authors should, at least, speculate on potential biological meaning of their results and, ideally, suggest a hypothetical model explaining their observations. Therefore, the Discussion should be expanded and rewritten correspondingly, in my opinion.”

Response: Thank you for the excellent suggestion. Based on your suggestion, we have conducted a thorough literature survey on the known functionalities of the candidate genes that might have mechanistic links with IR stress response. Indeed, we found all three genes, NCOA4, ATE1 and FGF22, have interesting functional relevance with IR stress response, especially with DNA damage response.

Specifically, NCOA4 (Nuclear Receptor Coactivator 4) is a transcriptional coactivator of nuclear hormone receptors, which could inhibit the activation of DNA replication origins (Mol Cell., 2014, PMID:24910095) and prevent replication stress, maintain genome integrity and reduce DNA damage (Cell Rep., 2022, PMID: 35977492).

ATE1 (arginyltransferase 1), a highly conserved gene across the eukaryotic domain, has been reported critical for suppressing the outcome of DNA mutagenesis during DNA-damaging stress (e.g. Nat Commun., 2023, PMID:36709327; Cell Death Dis., 2016, PMID:27685622). FGF22 (fibroblast growth factor 22) has also been reported to protect L02 cells from H₂O₂-induced oxidative damage via suppression of

mitochondrial apoptosis pathways (Protein Expr Purif., 2018, PMID:29627393). Taken together, the literature analyses on these genes suggest that there are strong mechanistic links between their altered m⁶A methylation levels and irradiation exposure. We have now added this content to the Discussion section of the revised manuscript (pages 25-26, lines 513-526).

Finally, thank you again for your time and effort, and for helping to improve the manuscript. We have highlighted the changes in the resubmitted manuscript. We hope that these changes have made it more appropriate for publication, and we look forward to your response.

REVIEWERS' COMMENTS

Reviewer #1 (Remarks to the Author):

Authors addressed (at least attempted to address) the comments from prior reviews.

Reviewer #2 (Remarks to the Author):

The authors have addressed all my comments. The quality of the manuscript is significantly improved now.

I have two minor suggestions:

I think that it may make sense to move the results from Fig. S4 to the main manuscript. In my opinion, they are rather important.

The text needs additional editing for typos and some grammatical errors.

Point-by-Point Responses

We'd like to thank the editor and both reviewers for their valuable comments. We greatly appreciate the input and have enclosed a revised version that would hopefully address all the concerns in the reviews. We detail our responses to the reviews below.

Reviewer #2.

Many thanks to Reviewer #2 for your suggestion of this manuscript. We've addressed these issues as follows:

“The authors have addressed all my comments. The quality of the manuscript is significantly improved now. I have two minor suggestions:”

1. “I think that it may make sense to move the results from Fig. S4 to the main manuscript. In my opinion, they are rather important.”

Response: Thank you for the excellent suggestion. We have moved most of the results (3 of the 4 m⁶A sites examined by SELECT) from **Fig. S4** to **Fig. 2k and l**. Due to the space limit of Fig. 2, the result of the last m⁶A site (A799) has been kept in Fig. S4.

2. “The text needs additional editing for typos and some grammatical errors.”

Response: We have made additional editing for typos and some grammatical errors in the manuscript.

Finally, thank you again for your time and effort, and for helping to improve the manuscript.